# Spatial fidelity of workers predicts collective response to disturbance in a social insect

James D. Crall [1], Nick Gravish [2], Andrew M. Mountcastle [3], Sarah D. Kocher [4], Robert L. Oppenheimer [5], Naomi E. Pierce [1] & Stacey A. Combes [6]

Individuals in social insect colonies cooperate to perform collective work. While colonies often respond to changing environmental conditions by flexibly reallocating workers to different tasks, the factors determining which workers switch and why are not well understood. Here, we use an automated tracking system to continuously monitor nest behavior and foraging activity of uniquely identified workers from entire bumble bee (*Bombus impatiens*) colonies foraging in a natural outdoor environment. We show that most foraging is performed by a small number of workers and that the intensity and distribution of foraging is actively regulated at the colony level in response to forager removal. By analyzing worker nest behavior before and after forager removal, we show that spatial fidelity of workers within the nest generates uneven interaction with relevant localized information sources, and predicts which workers initiate foraging after disturbance. Our results highlight the importance of spatial fidelity for structuring information flow and regulating collective behavior in social insect colonies.

[1] Department of Organismic and Evolutionary Biology, Harvard University, 26 Oxford St., Cambridge, MA 02143, USA. [2] Mechanical and Aerospace Engineering, University of California San Diego, Engineer Ln, San Diego, CA 92161, USA. [3] Department of Biology, Bates College, 2 Andrews Road, Lewiston, ME 04240, USA. [4] Lewis-Sigler Institute for Integrative Genomics, Princeton University, Princeton, NJ 08540, USA. [5] Department of Biological Sciences, University of New Hampshire, 105 Main St., Durham, NH 03824, USA. [6] Department of Neurobiology, Physiology, and Behavior, University of California Davis, Davis, CA 95616, USA. Correspondence and requests for materials should be addressed to J.D.C. (email: jcrall@oeb.harvard.edu)

Social insects (i.e., ants, bees, wasps, and termites) are among the most ecologically dominant and evolutionarily successful animals on the planet. Within social insect colonies, many individuals cooperate to perform crucial collective tasks, including foraging, caring for young, maintaining and cleaning the nest, and defending the colony from predators and social parasites[1–3].

Division of colony labor among workers is widely considered the key adaptation of social insects, with the specialization of workers on specific tasks theorized to improve colony performance[2], for example by reducing the costs of switching between tasks[4,5]. However, individual workers generally show flexibility in task performance[6–8], and social insect colonies are able to reallocate workers to different tasks when colony demands change[6,9–13]. Such flexibility is crucial for colony function, since the allocation of workers to tasks must change with either fluctuating resources or colony perturbations such as the loss of foragers to predation. In honey bees, for example, removal of older foragers leads to precocious development of younger nurses to replace forager losses[14]. Such flexible responses to disturbance are widespread in social insects (although not universal[15]) and often occur rapidly (e.g., within 2–3 min in the harvester ant *Pogonomyrmex barbatus*[16]).

Flexible reallocation of workers to tasks occurs in the absence of central control, through local interaction rules and information flow[17,18]. The sources of local information used in this distributed control of task allocation are diverse, and include tactile[19], visual[20], chemical[17,21], and acoustical[22] information exchanged in direct interaction with nestmates, or indirectly (for example through the nest structure[18]) in social insect colonies.

Although shifting colony labor demands are often filled by a non-random subset of colony workers[6,11], the factors driving particular workers to switch tasks over others are not well understood, particularly in species with less advanced social organization. Workers in social insect colonies show substantial inter-individual variation in many aspects of behavior, both within castes, as well as in species that lack clearly distinguishable worker castes[23–28]. While often described using different terminology[23], this variation is similar to what has been called animal "personality" (i.e., individually repeatable behaviors) outside of the social insects[29,30]. Individual behavioral variation can arise from multiple sources (reviewed for social insects in ref. [24]) and has important evolutionary and ecological consequences[31,32]. In social groups, the composition of different personalities plays a key role in determining group dynamics and collective behaviors[33–38].

One aspect of individual variation hypothesized to play a central role in the collective behavior of social insect colonies is sensitivity to task-specific stimuli (e.g., food reserves as a stimulus for foraging, or temperature as a stimulus for thermoregulation). Sensitivity to these stimuli—also known as response thresholds—may vary among individual workers, making certain workers more likely to perform particular tasks[39]. Previous theoretical work has shown that response thresholds can explain task specialization at the colony level[39,40], as well as worker flexibility in response to disturbance[40], consistent with empirical observations[6].

However, response threshold models have important limitations. First, empirical support for response thresholds is known for a only limited number of cases (such as nest thermoregulation behavior in bumble bees[41,42]), and results often suggest more complex dynamics than typically captured by models[43]. In addition, although not strictly required by response threshold models, a common simplifying assumption is that individuals are evenly mixed in space and time, with equal access to relevant task-specific stimuli.

Within social insect nests, however, the distribution of individuals is spatially heterogeneous[21,28,44–46]. Likewise, relevant information cues from sources such as food storage pots and developing young are heterogeneously distributed in space and time[18,47–49]. If individual workers vary in space-use, this is likely to generate variation in access to key information sources within the nest and play a role in structuring how individual workers perform particular colony tasks[48]. While some models explicitly address spatial fidelity within nests, these have most often treated spatial distributions as a consequence of age-based movements within the nest[50–52], mobility patterns[53], or aggression[54]. Despite growing interest in the spatial organization of work (and workers) in social insect colonies[46,48,55], the role of worker spatial fidelity in structuring access to local information sources within the colony and the importance of this aspect of worker spatial structure for task allocation is not well understood.

The role of individual behavioral variation in flexible task allocation in social insect colonies in general, and the functional role of worker spatial fidelity patterns in particular, thus remain largely unresolved. One challenge is that individuals vary in multiple phenotypic axes simultaneously: in addition to response thresholds and spatial fidelity, workers in social insect colonies can vary in morphology[56], physiology[11], activity level[10], aggression[57], and cognition[58], among others, all of which may play an important role in flexible task allocation. In addition, variation across multiple aspects of behavior is often correlated (i.e., behavioral syndromes[30]), making it difficult to parse which components play a functional role in colony dynamics.

A related challenge is that the processes giving rise to correlations between phenotypic traits and task performance may involve complex dynamics and feedbacks not revealed by examining static relationship between phenotype and task performance. For example, while in undisturbed colonies there may be a correlation between body size and foraging, this does not necessarily imply that body size will predict which workers switch to foraging after a perturbation (e.g., loss of foragers to predation). Performance of a task may also have dynamic feedbacks on other aspects of behavior. In honey bees, younger nurses generally lack the strong circadian activity patterns present in older foraging bees, since developing brood within the nest require round-the-clock care[59]. Circadian rhythms appear to be a consequence of foraging activity, as foragers that are artificially forced to revert to nursing behavior subsequently lose these circadian activity patterns[59]. While potentially widespread, such behavioral feedbacks are not well understood in the context of task allocation in social insect colonies.

Here, we use an automated, image-based tracking system (BEEtag[60]) to characterize behavioral variation across entire colonies of bumble bees (*Bombus impatiens*) and explore its role in flexible colony task allocation. Bumble bees live in relatively small (~50–200 workers) and simple (i.e., lacking clearly distinguishable worker castes) colonies. While there is evidence that certain colony tasks (such as foraging) are related to body mass[56], bumble bee colonies are characterized by weak division of labor[61] and do not display clear patterns of age-based polyethism[62] present in honey bees and many ants. While bumble bees lack the sophisticated communication of the honey bee waggle dance, several sources of local information flow affect foraging activity among workers, including olfactory and tactile cues gained from nestmates[63], information on colony nutritional status from food storage pots[18], and direct hunger signals from larvae[47].

We first test the hypothesis that bumble bee colonies actively respond to removal of foragers by flexible task reallocation among workers. We then examine the relationship between worker foraging activity and behavior within the nest. Specifically, we investigate (a) how the variation in worker nest behavior is

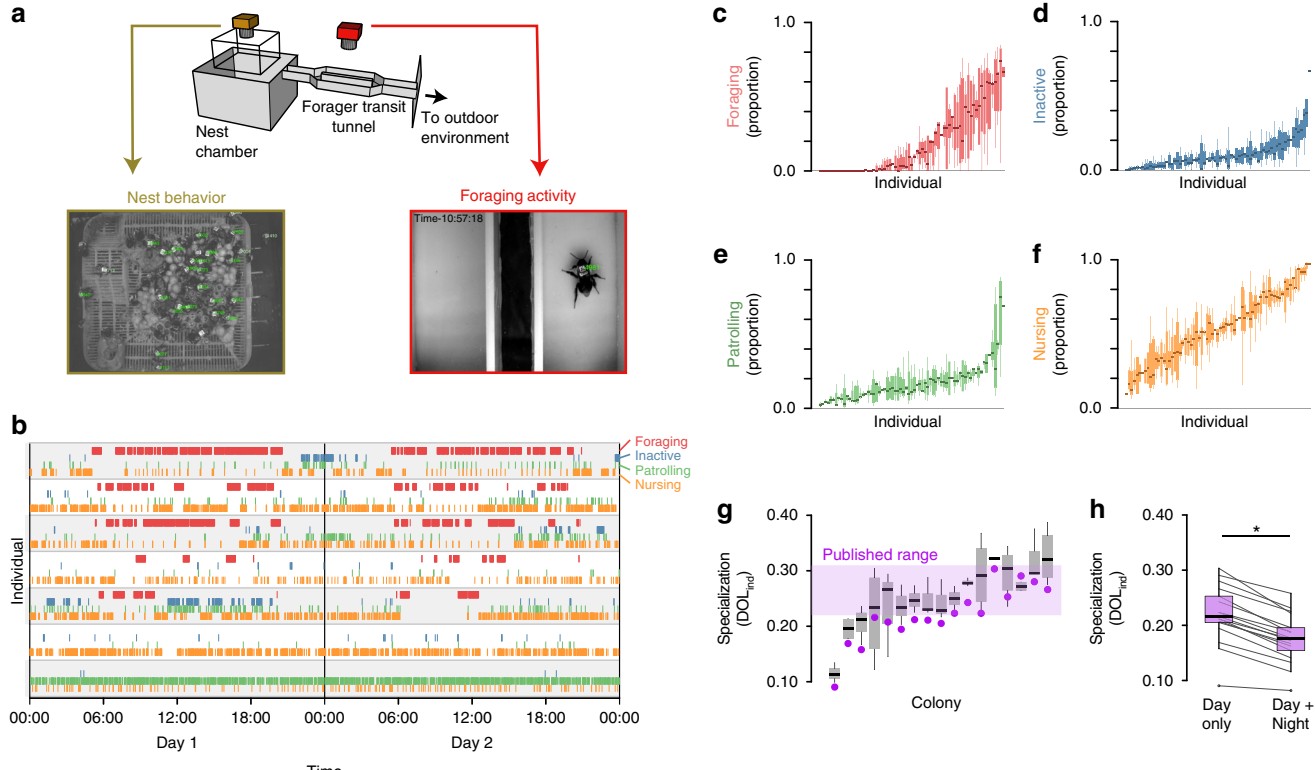

**Fig. 1** Automated behavioral tracking and task classification in bumble bee colonies. **a** Schematic diagram (above) and sample images from within the nest (left, orange) and the foraging tunnel (right, red), showing tracked individual BEEtags in green. **b** Representative traces of task performance from seven individual bees over 2 days from a single colony. Vertical lines indicate timesteps during which bees were foraging (F, red), inactive (I, blue), patrolling/cleaning (P, green), or nursing (N, orange). **c–f** Proportion of time spent engaged in different tasks by individual workers from the same representative colony. Boxplots in **c–f** show values for individual workers. Individuals are ordered in each plot according to the mean proportion of time spent performing that task. **g** Worker specialization (DOL$_{ind}$) values by colony. Filled purple markers show total estimates of each metric for each colony (i.e., combining data from all experimental days) and purple shaded box shows the range of published values across four colonies from the same species using manual task classification[61]. Gray boxplots show data when division of labor metrics are calculated for each experimental day separately. **h** Purple boxplots show total worker specialization values for each colony when only task performance from daytime hours is considered ("Day only"), vs. when round-the-clock data are included ("Day + Night"). Gray lines connect values for individual colonies. Throughout, boxplots show the median and inter-quartile range (IQR), while whiskers depict the data range (75th and 25th ±1.5*IQR, respectively)

organized (i.e., its correlation structure), (b) how it varies among individuals, and (c) the relationship between foraging activity and nest behavior in undisturbed colonies. We then test the hypotheses that (a) spatial fidelity of nest workers drives certain individuals to initiate foraging by biasing access to spatially localized nest information, and that (b) switching to foraging subsequently alters patterns of locomotor activity of workers within the nest.

## Results

**Collection of worker behavioral data.** We recorded 1.27 million nest behavior sequences and 26,511 foraging transits from 1717 individual *Bombus impatiens* workers, living in 19 colonies and foraging freely in the outdoor environment in Bedford, MA between July and October 2015 (Fig. 1a). For each colony, spatial locations and body orientations of uniquely identified workers within the nest were tracked regularly over 5-s intervals (Supplementary Movie 1) ~140 times daily (or about once every 10 min), 24 h per day, for up to 2 weeks, while foraging transits into and out of the nest were recorded with a motion-activated camera (Fig. 1a, Supplementary Movie 2).

**Automated classification of task performance.** We combined movement and location information of individual workers with spatial-mapping of key nest components (i.e., developing eggs, larvae, and pupae, wax pots for food storage, etc.; Supplementary Movie 3) to identify task performance of individual workers within the nest (Fig. 1b) at each time interval using four broad task groupings: (1) foraging, (2) nursing, (3) patrolling, or (4) inactivity. Automated classification of the three nest tasks (i.e., nursing, patrolling, and inactivity) had strong overall agreement with a human observer (86%, see Supplementary Table 2). The time each bee spent foraging was estimated by monitoring transits of individuals workers through the foraging tunnel connecting the nest chamber to the outdoor environments (Fig. 1a, see Methods for details). Individual bees frequently switched between tasks (Fig. 1b), with the vast majority (95%) of workers performing tasks related to at least three of the four tasks on a given day, and nearly all (99.7%) bees performing at least two different tasks each day.

Despite high levels of task flexibility, individual workers nonetheless showed strongly repeatable patterns of task propensity. Specifically, the amount of time spent on each individual task was repeatable from day to day, even though workers performed multiple tasks each day (Fig. 1c–f, one-way ANOVA; proportion of time nursing, df = 1132, F = 5.64, $p < 10^{-16}$; proportion of time foraging, df = 1132, F = 4.75, $p < 10^{-16}$; proportion of time inactive, df = 1132, F = 4.49, $p < 10^{-16}$; proportion of time patrolling, df = 1132, I = 5.17, $p < 10^{-16}$).

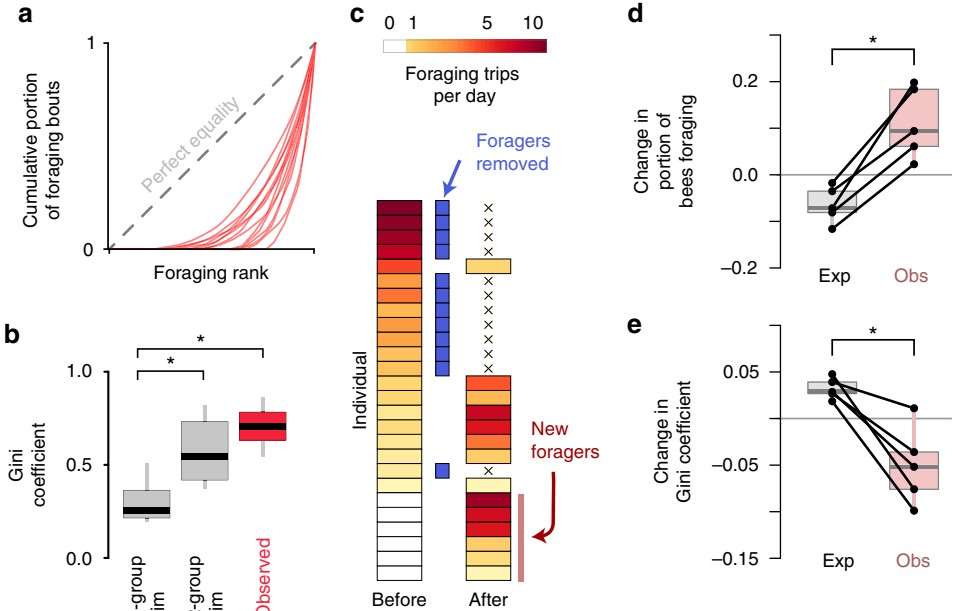

**Fig. 2** Foraging activity is regulated at the colony level. **a** Lorenz curves showing cumulative proportion of all foraging bouts (y axis) vs. foraging rank (x axis, sorted by the number of bouts per day performed by each worker) for individual colonies (n = 14, red lines). **b** Observed Gini coefficients across experimental colonies (red) vs. simulations in which observed foraging activity is randomly distributed across either all bees within colonies (1-group sim, gray) or across foragers and non-foragers (2-group sim, gray). **c** Foraging activity (mean foraging bouts per day) of individual bees from a single representative colony for the three days before and after an artificial disturbance (forager removal), where 12 foragers were removed from the colony (indicated by blue bars). Data for bees that performed no foraging before or after manipulation (n = 21) are not shown. **d, e** Expected (left) vs. observed (right) change in proportion of bees within the colony foraging (**d**) and in the Gini coefficient (**e**) after forager removal in five colonies. Black lines connect values from the same colonies before and after treatment. Boxplots show the median and inter-quartile range (IQR), while whiskers depict the data range. Expected values are based on the loss of removed foragers, given the null expectation that all remaining bees would continue their previous activity patterns

We quantified the specialization of workers to colony tasks ($DOL_{ind}$) using Gorelick's normalized mutual entropy method[64], and compared these values to published data for *B. impatiens*[61] (Fig. 1g, h). Worker specialization measured using data exclusively from during daylight hours was consistent with these published values[61] (Fig. 1g, h). In addition to calculating total worker specialization for each colony by pooling behavioral data across all experimental days, we calculated worker specialization values for each colony separately on each experimental day (Fig. 1g). Colonies showed variation in the degree of worker specialization that was stable across days (Fig. 1g; $DOL_{ind}$, one-way ANOVA, df = 16, F = 4.50, $p = 8.9 \times 10^{-7}$) and not related to colony size ($DOL_{ind}$ vs. colony size, linear regression, df = 15, t = −0.031 p = 0.98).

To assess the importance of continuous (i.e., round-the-clock) behavioral tracking for quantifying division of labor, we also calculated total worker specialization scores separately for each colony while either excluding overnight task performance data (8 p.m.–6 a.m., Fig. 1h "Day only"), or including these data (Fig. 1h, "Day + Night"). Including overnight behavioral significantly reduced worker specialization values across colonies (Fig. 1h).

**Distribution and regulation of colony foraging activity.** Analyzing individual foraging activity revealed that all colonies had a significant skew in the distribution of foraging activity (defined as the number of foraging bouts, rather than proportion of time) among workers, with the majority of foraging bouts performed by a relatively small number of bees (Fig. 2a). We quantified the statistical inequality in foraging activity using the Gini coefficient (Fig. 2a, b), for which values increasing from 0 to 1 reflect growing inequality among individuals, and found a significant

skew in foraging behavior within colonies (Gini = 0.71 ± 0.09). Observed Gini coefficients were higher than simulated permutations of foraging activity, treating either (a) all bees in the colony as a single group displaying equal foraging effort (Fig. 2b, df = 13, $t = -18.8$, $p < 10^{-10}$, paired t-test), or (b) workers as belonging to one of two equivalent groups, foragers or non-foragers (Fig. 2b, df = 13, t = −6.3, $p = 2.8 \times 10^{-5}$, paired t-test).

To investigate the regulation of foraging activity at the colony level, we artificially disturbed colonies by removing foragers (Fig. 2c) and tracking foraging activity of the remaining workers in the three days before and after experimental manipulation (Fig. 2c–e) in five separate colonies. The proportion of bees foraging after manipulation was significantly higher than expected based on the foragers lost, relative to the null expectation that all remaining bees would continue their previous activity patterns (Fig. 2d, paired t-test, t = 4.8, df = 4, p = 0.0083). A similar pattern held for the inequality of foraging activity, with observed Gini coefficients significantly lower than expected after manipulation (Fig. 2e, paired t-test, t = −4.0, df = 4, p = 0.02), as compared to the null expectation that only the remaining bees would maintain consistent foraging activity after manipulation. Neither the proportion of bees foraging nor the Gini coefficient of foraging activity were significantly shifted by a similarly timed manipulation where workers were temporarily removed, separately housed, and then replaced in the colony (Supplementary Fig. 1) and analyzed on the same time scale, suggesting that these changes in foraging activity are not due to fluctuations in foraging activity under natural colony dynamics. The increase in foraging activity after disturbance is unlikely to result from reduced colony size, as there was not a significant relationship between colony size and the proportion of bees foraging (linear model, df = 16, t = −0.39, p = 0.705).

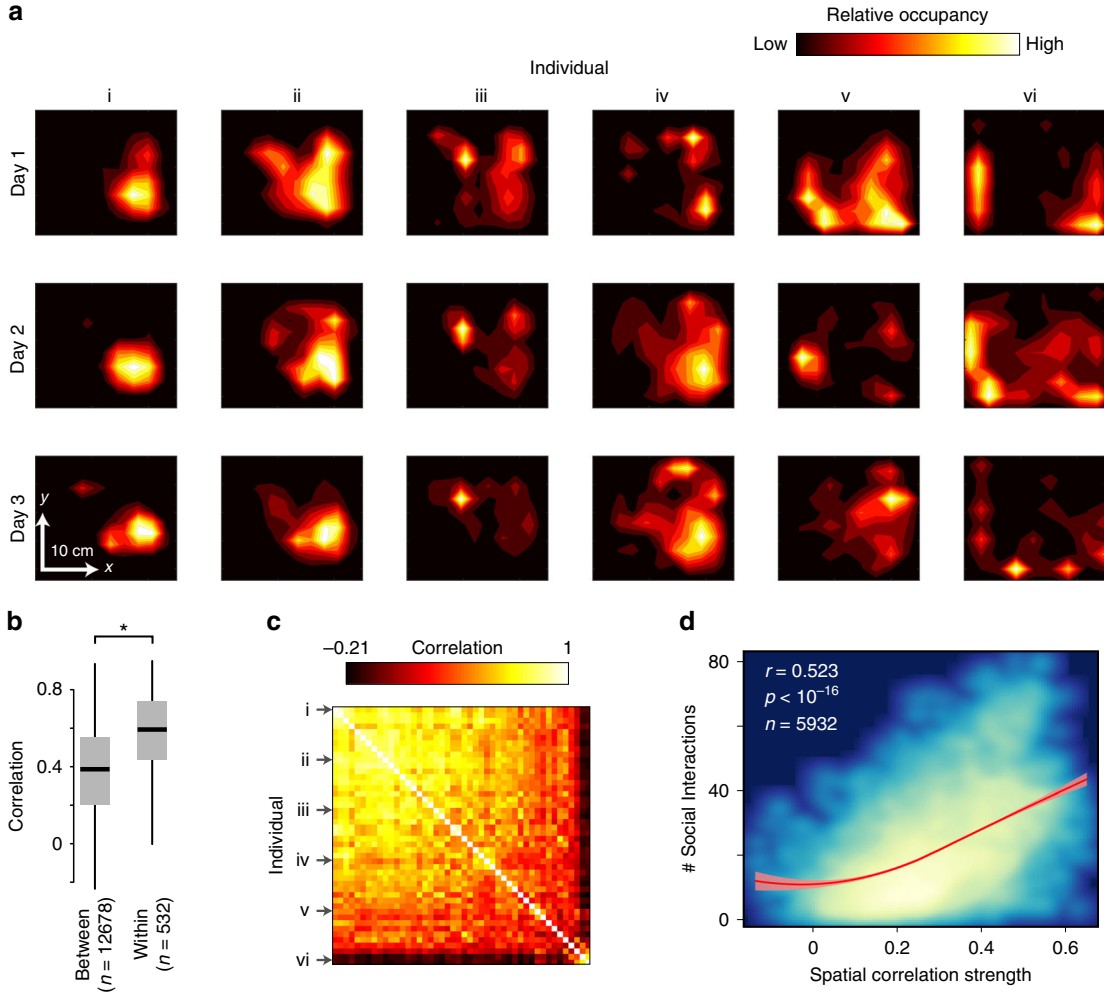

**Fig. 3** Workers show variable and repeatable patterns of spatial fidelity within the nest. **a** Representative spatial occupancy patterns within the nest of six bumble bee workers (i–vi) over three days from a single colony. Color indicates relative occupancy, smoothed by kernel density estimation. **b** Correlation of spatial occupancy within and between individuals across days from the same colony. Boxplots show the median and inter-quartile range (IQR), while whiskers depict the data range (75th and 25th ± 1.5*IQR, respectively). **c** Correlation matrix of spatial occupancy patterns between all workers for a single day from a single colony (same as in Fig. 1). Rows corresponding to the six individual workers (i–vi) shown in **a** are indicated by arrows. Color indicates correlation strength. **d** Relationship between mean spatial correlation strength to other bees (as in **c**) and the number of unique social interactions engaged in by that individual, across all days and colonies. Color indicates density of points, from blue to light yellow. Spearman correlation coefficient = 0.523. Thick red line shows local smoothing estimate (Loess) and ±95% CI (in light red)

**Data-driven characterization of nest behavior**. To provide a more complete description of worker behavioral variation than the task classification approach described above, we generated a high-dimensional data set of worker activity level, movement patterns, and spatial distribution within the nest. We found that individual workers have highly variable patterns of spatial occupancy within the nest, which were consistent across days (Fig. 3a, b, $p < 10^{-39}$ for all colonies, two sample $t$-test). We calculated pairwise correlations between spatial occupancy patterns of individual bees with each nestmate as a proxy for social information flow within the nest (Fig. 3c). This metric incorporates rates of both direct physical interaction with nestmates (i.e., workers with high spatial correlation strength scores interaction with more unique nestmates, Fig. 3d), as well as indirect information exchange via substrate-based and other stigmergic interactions[49]. Some workers had spatial occupancy patterns strongly correlated with nestmates (e.g., individual "i" in Fig. 3a, c), while others had weak spatial correlation to nestmates (e.g., individual "vi" in Fig. 3a, c). We calculated the mean spatial correlation strength of each bee to all of its nestmates ("Spatial correlation

strength" in Fig. 3d), hereafter referred to as "social interaction strength". For each worker, we also estimated rates of interaction with both developing brood ("Brood interaction rate" below) and food storage pots ("Waxpot interaction rate") within the nest.

We combined these interaction rates with other metrics of space-use, locomotor patterns, circadian activity, and dispersal within the nest to generate a data set of 23 metrics of nest behavior. We then performed a principal components analysis to reduce the dimensionality of this data set (Fig. 4, see Supplementary Table 1 for a list of variable descriptions and principal component loadings, and Supplementary Fig. 2 for the correlation matrix of nest behavior metrics).

The first two principal components explained a combined 56% of the observed variation in our high-dimensional data set of worker behavior in the nest (hereafter, "nest behaviors", Fig. 4). The first principal component of nest behavior (PC1, Fig. 4a, b) was correlated strongly with aspects of spatial occupancy and was higher for bees that were more centrally located within the nest, with higher PC1 scores associated with shorter distances from the nest center, smaller occupancy ranges, and increased rates of

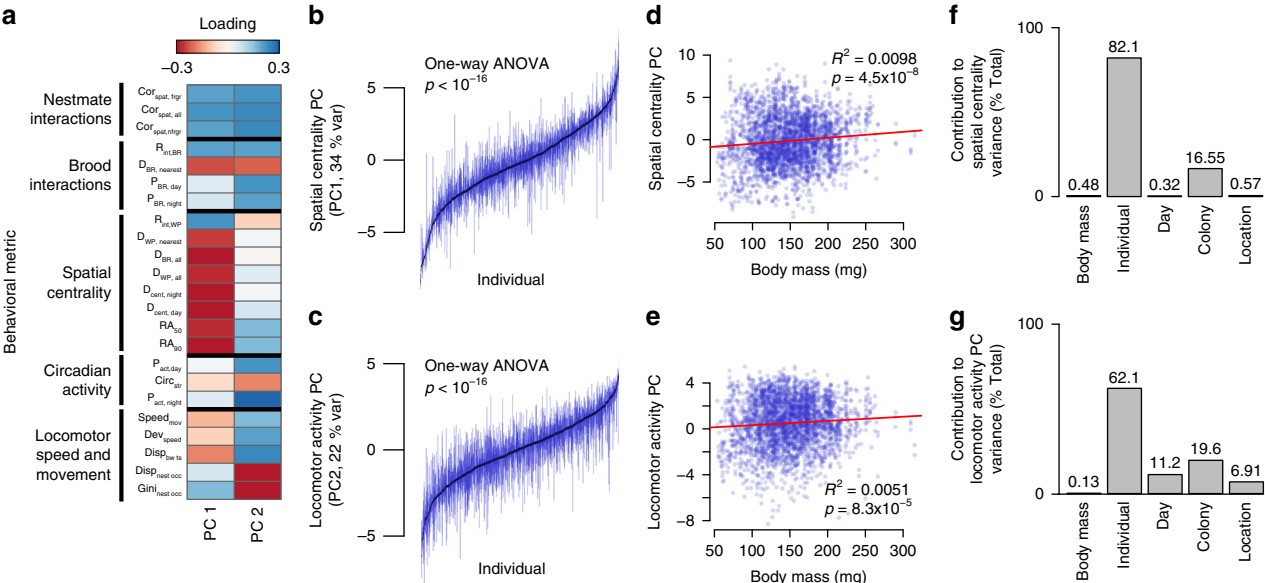

**Fig. 4** Bumble bee workers show strong individual variation in nest behavior that is stable across days and not associated with body size. **a** Loadings of individual behavioral metrics on Principal Components 1 and 2, with color indicating direction and strength of loading. Labels indicate clusters of correlated variables (Supplementary Fig. 3). **b,c** PC1 ("Spatial centrality" (**b**)) and PC2 ("Locomotor activity" (**c**)) scores across days for all individuals. Boxplots show the median (black marker) and inter-quartile range (IQR, gray bars) for each worker's score across days, with individuals arranged according to median values. **c,d** Relationship between body mass and PC1 (**d**) and PC2 (**e**). Transparent filled markers show data for individual workers on different days, and $R^2$ and $p$ values are calculated from a linear regression (red line). **f,g** Relative contribution of body size (mass, mg), individual, day, colony, and experimental location to variance in PC1 (**f**) and PC2 (**g**) scores in undisturbed colonies. Values represent the relative contribution (% of explained variance) using hierarchical partitioning

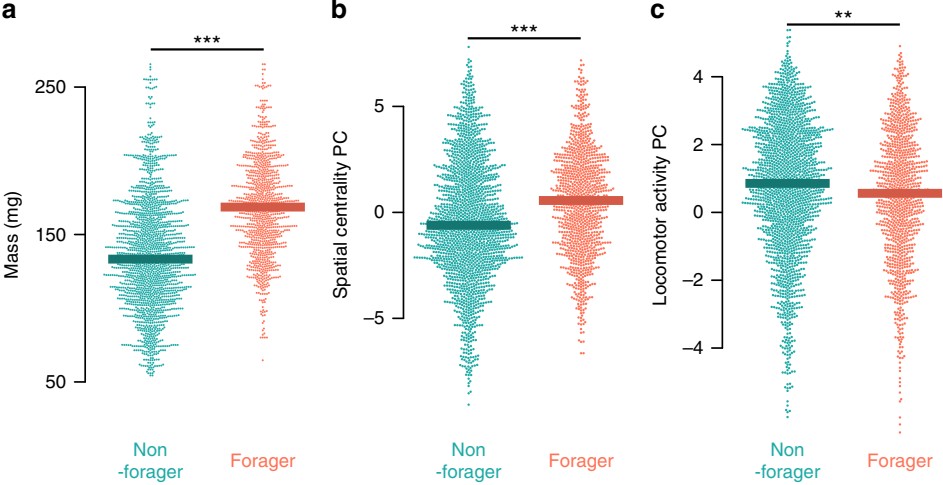

**Fig. 5** Foraging is correlated with body mass and nest behavior in undisturbed colonies. Beeswarm plots of body mass (mg, **a**), spatial centrality score (**b**, PC1), and locomotor activity score (**c**, PC2) by foraging status (forager vs. non-forager) in undisturbed colonies. Filled points show data for individual workers on unique days, and thick lines show median values for each group. **\*\***$p < 0.005$, and **\*\***$p < 0.0005$ in a generalized linear mixed-effects model

interaction with developing brood, food storage pots, and nestmates, among others (Fig. 4a, b).

In contrast, the second principal component of nest behavior (PC2, Fig. 4) had stronger loading from metrics of mobility patterns ("Locomotor speed and movement patterns" and "Circadian activity" metrics in Fig. 4) and was higher in bees that tended to be more mobile. Specifically, higher PC2 scores were associated with higher proportions of time spent mobile and instantaneous movement speeds, greater long-term spatial displacement within the nest, and decreased circadian activity scope (i.e., smaller differences between the proportion of time spent active during the day and at night).

Both PC1 and PC2 had strong loading from a suite of behavioral metrics associated with social interactivity (Fig. 4a), including spatial correlation strength to nestmates, as well as proximity to and proportion of time spent on the brood, rather than on the waxpots. Higher PC1 scores (i.e., more physically central) and higher PC2 scores (i.e., higher locomotor activity level) were both associated with increased interaction with nestmates and brood (Fig. 4a, Supplementary Fig. 2).

Individual workers showed significant repeatability for both principal components of nest behavior across days (Fig. 4b, c, PC1, one-way ANOVA, df = 1297, $F = 5.5$, $p < 10^{-16}$; PC2, one-way ANOVA, df = 1297, $F = 2.2$, $p < 10^{-16}$). Hereafter, we use the

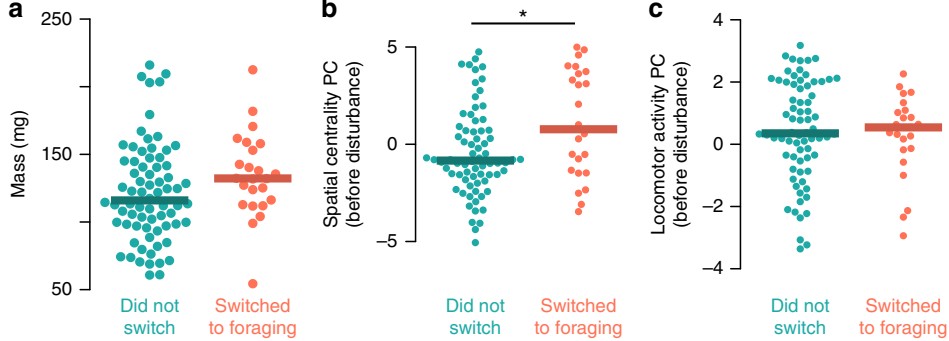

**Fig. 6** Spatial centrality before disturbance predicts foraging initiation after disturbance. Beeswarm plots of body mass (mg, **a**), spatial centrality score (**b**, PC1), and locomotor activity score (**c**, PC2) by whether workers initiated foraging after disturbance, among workers that performed no foraging before disturbance. Filled points show mean values for individual workers in the three days before disturbance, and thick lines indicate median values. *$p < 0.05$ in a generalized linear mixed-effects model

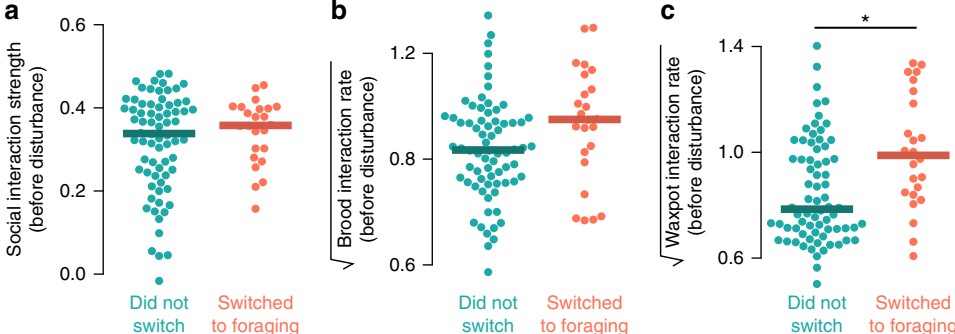

**Fig. 7** Interaction with food storage pots predicts foraging initiation after disturbance. Beeswarm plots of nest behavior metrics before experimental disturbance by foraging status after disturbance, among bees that had performed no foraging before disturbance. **a** Social interaction strength (estimated by mean spatial correlation strength to nestmates, see above). **b** Rate of interaction with brood, and **c** rate of interaction with waxpots where food may be stored. Points show values for individual bees, and thick lines show median values for group. *$p < 0.05$ in a generalized linear mixed-effects model

terms "Spatial Centrality PC" and "Locomotor Activity PC" in place of "PC1" and "PC2", respectively, for simplicity. Both Spatial Centrality PC and Locomotor Activity PC scores had a very weak (while statistically significant, Fig. 4d, e) relationship with body size, explaining less than 1% of variance in either principal component (Fig. 4d–g). The majority of variation in nest behavior in both principal components appears to result from stable variation among individuals that is not strongly related to body size, experimental day, colony, or experimental location (Fig. 4f-g).

**Nest behavior and flexible task allocation**. We next examined the relationship between nest behavior and foraging status in both undisturbed colonies, and in colonies responding to experimental forager removal. First, we examined the relationship between principal components of nest behavior, body mass, and foraging activity in workers from colonies before experimental disturbance (Fig. 5). Consistent with previous results in bumble bees[56], we found strong evidence that larger bees were more likely to forage (Fig. 5a, generalized linear mixed-effects model, df = 3039, z = 12.16, $p < 10^{-16}$). Accounting for the effects of body size, however, we found that higher Spatial Centrality PC scores and lower within-nest Locomotor Activity PC scores were also both significantly correlated with foraging activity in undisturbed colonies (Fig. 5b, c; generalized linear mixed-effects model, Spatial Centrality PC (PC1), df = 3039, z = −4.16, $p = 3.2 \times 10^{-5}$; Locomotor Activity PC (PC2), z = −4.64, $p = 3.6 \times 10^{-6}$).

To examine whether variation in worker behavior and in the nest before disturbance predicts which workers will switch tasks after disturbance, we tested the relationships between worker nest behavior prior to forager removal (as described above) and the probability of switching to foraging after manipulation. We focused exclusively on bees that had performed no foraging in the three days prior to disturbance (Fig. 6). Body size did not significantly predict whether workers initiated foraging after disturbance (Fig. 6a, generalized linear mixed-effects model, df = 88, z = 1.29, p = 0.197). In contrast, Spatial Centrality PC scores significantly predicted which workers initiate foraging in response to disturbance (Fig. 6b, generalized linear mixed-effects model, df = 88, z = 2.36, p = 0.018). Locomotor Activity PC score had no effect on the probability of initiating foraging (Fig. 6c, generalized linear mixed-effects model, df = 88, z = −0.09, p = 0.93).

To examine the specific components of spatial variation that drive nest workers to initiate foraging, we tested the relationship between three behavioral metrics (social interaction strength, brood interaction rate, and waxpot interaction rate) of workers before disturbance and forager switching status, again among workers that performed no foraging before disturbance (Fig. 7). These metrics were selected because they had strong loading on the Spatial Centrality PC principal component (Fig. 4), clustered independently (Supplementary Fig. 2), and are related to known information cues within the nest (social interactions[63], larval hunger signals[47], and food storage information[18] for social interaction strength, brood interaction rate, and waxpot interaction rate, respectively). Waxpot interaction rate significantly predicted which workers initiated foraging after disturbance (Fig. 7a, generalized linear mixed effects model, df = 88, z = 2.48, p = 0.013), but social interaction strength and brood interaction rate did not (Fig. 7b, c; generalized linear mixed effects model:

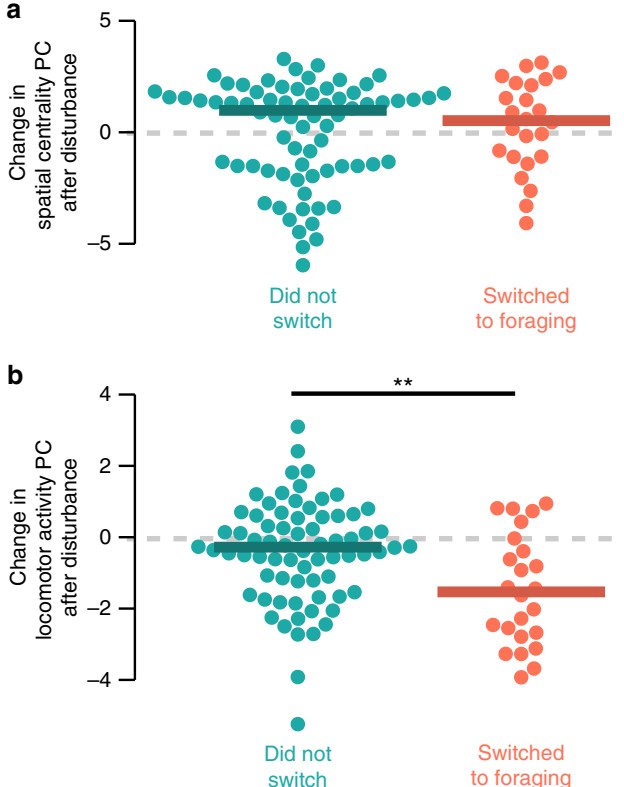

**Fig. 8** Initiating foraging decreases locomotor activity within the nest. Beeswarm plots of change in nest behavior metrics after disturbance by foraging status after disturbance, among bees that had performed no foraging before disturbance. **a** Change in spatial centrality (PC1) scores, and **b** change in locomotor activity scores (PC2). Points show mean values for individual bees during the three days after disturbance, and thick lines show median values for group. **p < 0.005 in a linear mixed-effects model

social interaction strength, df = 88, $z = 1.16$, $p = 0.25$; brood interaction rate, df = 88, $z = 0.34$, $p = 0.74$).

Next, we tested whether switching to foraging affected other aspects of within-nest behavior during the three days after the disturbance occurred, again among bees that were not foraging before disturbance. We found that initiating foraging subsequently reduced Locomotor Activity PC scores (Fig. 8b, linear mixed effects model: change in PC2, df = 92.9, $z = -3.120$, $p = 0.0019$), but had no detectable effect on Spatial Centrality PC scores (Fig. 8a, linear mixed effects model: change in PC1, df = 92.6, $z = 1.45$, $p = 0.15$). We found qualitatively similar patterns when individual metrics of nest behavior were tested separately (Supplementary Fig. 3).

## Discussion

Despite growing interest in spatial dynamics within social insect colonies[21,45,55,65], the importance of spatial fidelity for flexible task reallocation is not well understood. Our results provide evidence for three key tenets supporting a functional role of spatial fidelity in flexible task allocation in bumble bees: (a) the existence of spatial fidelity among workers (Fig. 3), (b) a correlation between spatial fidelity and relevant information sources within the colony (i.e., interactions with nestmates, Fig. 3, and interactions with brood and waxpots, Fig. 4, Supplementary Fig. 4), and (c) a predictive role of spatial occupancy in determining which individuals respond to changing colony labor demands (Figs. 6, 7).

Our results provide clear evidence for stable, individually-specific patterns of spatial occupancy within bumble bee colonies; spatial distributions of bumble bees within the nest varied between individuals, but were significantly repeatable within individuals across days in all colonies examined (Fig. 3). Previous work showed that over extended time scales (with each worker's location measured once a day for up to 49 days), a subset of workers (~20%) had spatial fidelity zones either larger or smaller than expected under null simulations[44]. Our results show that spatial fidelity patterns are strongly repeatable across individual days, with workers returning to the same spatial zones within the nest in predictable patterns each day (Fig. 3).

We also show that spatial distributions are correlated with important sources of information within the nest, supporting a functional role in patterning information flow within insect colonies[48]. Multiple sources of information are known to affect worker behavior in bumble bee colonies, including social information[63,66], signals from developing brood[47], and information stored in food pots[18]. Our results show that an individual bumble bee's spatial fidelity patterns within the nest are correlated with rates of interaction with all three of these relevant information sources (Figs. 3, 4, S2, S4).

Finally, our results provide evidence for a predictive role of spatial fidelity in determining task allocation. In addition to being correlated with foraging activity in undisturbed colonies (Fig. 5b), a worker's spatial occupancy within the nest predicts whether it will initiate foraging when colony-level demand for foraging is increased via artificial disturbance (Fig. 6b). While other phenotypic traits (i.e, body mass and locomotor activity) are significantly correlated with foraging activity (Fig. 5a, c), neither of these predict foraging initiation after disturbance (Fig. 6a, c).

Our results also suggest that, while information from multiple sources is largely correlated (bees with higher Spatial Centrality PC scores had more unique physical interactions with nestmates, Fig. 3d, as well increased rates of interaction with both developing brood and waxpots Fig. 4a, S4), each source of information does not contribute equally to flexible task allocation. Specifically, spatial overlap with food storage pots before disturbance was the strongest independent predictor of which workers initiated foraging after disturbance. This suggests that information stored in waxpots, rather than hunger signals from the larvae and/or interactions with nestmates, is the most important signal workers use to initiate foraging in response to disturbance of the colony's foraging workforce.

We also found that while spatial occupancy is a significant predictor of which workers begin foraging in response to disturbance (Figs. 6b and 7), this change in task performance does not significantly affect spatial occupancy patterns within the nest (Fig. 8a, Supplementary Fig. 3). At least in the case of bumble bees, spatial occupancy thus appears to be a precursor, rather than a product, of foraging. What gives rise to this individual variation in spatial occupancy is unclear, however, and the ontogeny of spatial fidelity is an important avenue for future study.

Previous work has suggested that individual differences in mobility could explain patterns of workers' space-use within social insect colonies[44,53], for example with more mobile bees diffusing to the nest periphery and thus driving patterns of spatial distribution and task allocation[53]. Our results do not support such a link: patterns of space use and locomotor activity are largely independent (Fig. 4, Supplementary Fig. 2). To the extent that mobility and space use are linked in the context of foraging, the relationship appears to be the opposite; rather than variation in worker mobility driving patterns of space use, spatial fidelity impacts foraging activity, which in turn can affect locomotor activity (Figs. 5–8).

Together, our results support a functional, predictive role for variable spatial fidelity among workers in determining the dynamics of flexible task allocation in bumble bee colonies. Within the framework of response threshold models, the probability of performing a given task is a product of not only the individual's response threshold, but also the stimulus level for that task perceived by the individual worker (rather than the absolute, colony-wide stimulus level per se). While response threshold models typically assume, for example, that these stimuli are evenly distributed within colonies, in reality local cues and signals relevant for collective behavior are distributed heterogeneously[48]. Our results highlight the importance of investigating not only intrinsic individual differences in task preferences (for example in the form of variable individual response thresholds), but also the idiosyncratic perceptual worlds inhabited by unique individuals and the factors that drive this variation, including spatial fidelity. These mechanisms are not exclusive, however; spatial fidelity could either mitigate or exacerbate inter-individual variance from response thresholds, and the interaction between these two factors is an important direction for future study.

In contrast to spatial fidelity, we found evidence that locomotor activity within the nest is affected by initiating foraging, rather than predicting patterns of task switching. In undisturbed colonies, foragers had lower Locomotor Activity PC scores than non-foragers (Fig. 5c), but Locomotor Activity PC score did not predict which bees initiated foraging activity after disturbance (Fig. 6c). Rather, workers that had not previously foraged but initiated foraging after disturbance subsequently showed reduced Locomotor Activity PC (Fig. 8b, Supplementary Fig. 3). A similar pattern has been shown for the regulation of circadian rhythm in honey bees in response to shifting from foraging to nursing[59], and may reflect the high physiological demands of foraging.

We found evidence that the overall intensity and distribution of foraging activity among workers are regulated at the colony level in bumble bees (Fig. 2). Consistent with previous findings in honey bees[8] and bumble bees[67,68], individual workers showed significant repeatability in foraging behavior (both between foragers and non-foragers, and between foragers, Fig. 2b) and the majority of foraging bouts were performed by a minority of workers. In response to simulated predation (via forager removal), colonies increased foraging activity and decreased inequality (i.e., Gini coefficients) relative to predicted effects. While active responses to forager removal are well known from other social insect species[6,8] (but not universal[15]), to our knowledge similar responses to forager removal have not been shown before for bumble bees. Factors determining the shape and regulation of this distribution will require additional research, but it's possible that they arise from variation in within-colony task propensities (regulated either by spatial fidelity or response-thresholds).

While our results suggest that bumble bee colonies actively respond to the loss of foragers, it remains unclear whether similar effects would be seen by removal of different sets of workers within the colony. For example, recent work has shown that *Temnothorax* ants actively replace workers that are actively engaged in colony tasks, but not "inactive" workers[10]. Bumblebee workers respond to removal of nest workers by increasing rates of larval feeding[69], but it is unknown how the colonies responds to removal of workers performing other colony tasks, or a random subset of workers. While spatial fidelity—and the rate of interaction with waxpots in particular—predicts which workers switch to foraging after removal of the colony's more active foragers, it remains entirely unknown whether spatial fidelity plays a similar role for colony responses to other disturbance types.

Our results highlight the ubiquity and importance of individual variation in behavior among workers in bumble bee colonies that is not associated with morphology. Given the strong variation in worker body size in bumble bees, previous work has focused on behavioral variation associated with body size (i.e., alloethism[56,70,71]). Consistent with previous work[56], we find a significant (if only moderately strong) relationship between body size and foraging among *B. impatiens* workers (Fig. 5a). However, body size has only a very weak relationship with behavior of workers within the nest (explaining <1% of variance in both principal components, Fig. 4d–g). Rather, even after accounting for variation in behavior associated with several other factors (e.g., variation between colonies, across days, and experimental locations), the dominant factor explaining variation in nest behavior is strong and temporally stable individual variation, the origin of which in bumble bees is largely unknown.

Our results also demonstrate the potential of automated tracking methods to deepen our understanding of division of labor and task allocation in social insect colonies. We show that automated classification using a relatively simplistic task classification system can recapture patterns of colony-level worker specialization consistent with traditional, manual behavioral classification (Fig. 1g). We also show that continuous monitoring of task performance and behavior, which would be difficult or impossible using manual techniques, is critical for fully describing colony division of labor; performing behavioral monitoring only during the day systematically overestimates the degree of task specialization among workers (Fig. 1h), likely a consequence of the circadian structure of task performance at the individual and colony level. We also demonstrate the utility of these scalable techniques to address novel questions, such as the variation in colony division of labor across time or between colonies (Fig. 1g).

Flexible responses to changing labor demands are critical to colony survival in response to ecological perturbations (e.g., fluctuating resource availability, changes in the abiotic environment, or biotic interactions such as disease or predation) and are key to the ecological and evolutionary success of social insects[72]. Our findings highlight both the ubiquity of behavioral heterogeneity within social insect colonies and the functional role this variation plays in colony resilience. Such flexibility could be crucial for bumble bees, which have small colonies (~50–200 workers, compared to 40,000 or more workers in honey bees, or up to hundreds of thousands of workers in ant colonies), and often live in challenging and volatile alpine environments[73–75].

Gaining insight into the ecological and evolutionary forces driving behavioral diversity in social insect colonies requires a deeper understanding of the extent and function of variation between workers within colonies, across species and under relevant environmental conditions. Automated tracking techniques and the large, multidimensional behavioral datasets they generate can provide a rich and increasingly nuanced description of behavior across entire colonies facing relevant environmental challenges. These techniques may thus prove crucial in elucidating traits driving patterns of task allocation and their functional importance for the ecology and evolution of social insects.

## Methods

**Animal care and tag attachment**. We obtained 19 bumble bee (*B. impatiens*) colonies (112.7 ± 38.5 workers) from BioBest between July 13th and October 7th, 2015. Before beginning behavioral experiments, all bees (including the queen) were removed from the colony and cold-anaesthetized at 4 °C. All brood and structural nest components were transferred to a custom-designed nest chamber (see "Tracking Arena" below). Each bee was outfitted with a unique BEEtag printed on water-resistant paper, attached with cyanoacrylate glue to the mesoscutum, and body mass was recorded (to 0.1 mg precision). Bees were then returned to the tracking chamber and the colony was placed in one of three identical tracking arenas (Fig. 1a).

**Tracking arena**. Three custom-designed, identical tracking arenas (Fig. 1a, $0.20 \times 0.19 \times 0.13$ m) were used. The walls and floor of the nest chamber were constructed from laser-cut black acrylic, with a clear acrylic top. A monochrome digital camera (DMK 24UJ003, USB 3.0, Imaging Source, $3856 \times 2764$ pixels) with a wide-angle lens (Fujinon, 2.8–8 mm) was mounted on aluminum construction rails (25 mm, Thorlabs) above the clear top of the nest box. The nest was illuminated with red LED panels (Knema Lighting), which caused minimal disturbance to nest behavior since bees have very poor sensitivity to red light. The nest box, camera, and lighting array were covered with black cloth to exclude ambient light.

Each nest chamber was located in a temperature-controlled indoor environment, with direct access to the outdoors via a custom-designed foraging tunnel (Fig. 1a). The foraging tunnel was constructed from acrylic (3.1 mm thick), with opaque-white on the floor and sides of the tunnel, and clear on the top to allow for imaging. A digital camera (PointGrey Chameleon3, $1288 \times 964$ pixels) was mounted above the parallel middle sections of the foraging tunnel. This region of the foraging tunnel was monitored using a single red LED array identical to those above the nest chamber on a 16:8 h light:dark cycle.

**Colony deployment and experimental treatments**. Each colony was deployed to one of three locations at the Concord Field Station (Bedford, MA, all within 200 m of each other) for 14–21 days and automated tracking of nest and foraging behavior was initiated within 24 h of colony deployment. After tagging, each colony was supplied with a small amount of nectar (BioGluc) and fresh pollen (Koppert), after which no additional food was supplied and colonies were forced to forage in the outdoor environments for both nectar and pollen. All colonies initiated foraging within 24 h.

After establishment, nest and foraging data were recorded continuously, except on days when experimental manipulations were performed, which varied between colonies. Five colonies were subjected to experimental manipulations described below, and data collected before and after manipulations were analyzed. Fourteen colonies were subjected to experimental manipulations not reported here (although see Supplementary Fig. 1) in the analyses.

**Image acquisition**. In all three behavioral arenas, video data from both nest and foraging cameras were acquired directly to a PC Desktop computer using custom Matlab scripts. For nest behavior, video frames were captured at 2 Hz for 5 s (10 frames total), saved to an external hard drive, and immediately processed using BEEtag[60] (Supplementary Movie 1). After processing was complete (~2–5 min), data collection was again initiated. Behavioral sequences were collected from each nest ~140 times daily (or about once every 10 min; Supplementary Movie 1), 24 h per day, for up to two weeks.

Foraging transits were recorded via motion capture by the foraging camera using custom Matlab scripts (Supplementary Movie 2). With both channels of the foraging tunnel in view, images were recorded whenever motion was detected within the camera's field of view and written directly to an external hard drive. At 22:30 each evening, after foraging had ceased for the day, motion capture image collection ceased and that day's foraging images were processed using BEEtag[60]. Image collection began again after all images had been processed (always before 6 a.m. the following morning).

**Post processing of tracking data**. The BEEtag software records the location, orientation, and identity of any BEEtags located within a single frame[60] from either the nest or foraging cameras. To conservatively ensure that data from bees that had died within the colony were not included in the analysis, we ignored all data after the last four instances of observable movement of individual tags. After this initial pass, two-dimensional spatial coordinates of each tag were scaled and corrected for lens distortion using the Camera Calibration toolbox in Matlab. Missing coordinate "holes" within each nest video sequence were filled using linear interpolation.

**Spatial mapping of nest structures**. We manually mapped nest elements for each day and each colony using a custom Matlab script (Supplementary Movie 3). This script allowed manual mapping of the location of all brood (eggs, larvae, and pupae) and waxpots. On days of experimental manipulations, forager removal, or when the nest was otherwise physically disturbed for any reason, nest elements were not mapped and nest behavior was not analyzed.

**Analysis of foraging behavior**. We used tag-tracking data from the camera over the foraging tunnel to estimate the foraging activity of individual bees (Fig. 1a), assessing the timing and movement direction of individual bees using their tag orientation output from the BEEtag tracking software (Supplementary Movie 2).

From these data on time, identity, and movement direction of tags from the foraging camera, we estimated foraging activity as the number of unique foraging transits performed by each bee. We removed any foraging transits separated by <3 min from previous transits (although including shorter trips had no qualitative effect on any qualitative patterns reported here, over a range of time thresholds tested). While some movements in and out of the colony could be associated with nest defense (rather than foraging), these defensive behaviors can be easily distinguished visually by bees clustering at or near the nest entrance, and were rarely observed in this data set.

**Analysis of behavior within the nest**. For every bee identified, we estimated multiple components of task performance and nest behavior for each 5-s video sequence (or "timestep") separately for each bee, for each day of recording. For each timestep, we measured movement speed as the median of instantaneous frame-to-frame speed within a single video sequence. Movement speeds below a threshold of $10^{-4.3}$ m/s per second were considered to be stationary, based on the bimodal distribution of instantaneous movement speeds, the lower mode of which was assumed to result from noise in digital tracking (Supplementary Fig. 5).

**Task performance**. We used the spatial association with nest elements, in combination with movement information, to group the behavior of individual workers at each timestep into one of four behavioral clusters: foraging, nursing, patrolling, or inactive. Bees were considered to be foraging for time spans between recorded exits from the colony and subsequent entrances, unless they were located within the nest during that period (to compensate for times when foragers may be "missed" by the foraging camera, e.g., from bees transiting the foraging tunnel on the side wall, which occurred rarely). Bees within the nest that were physically associated (i.e., within 1 cm) with eggs, larvae, pupae, or waxpots were considered to be nursing. Bees that were not associated with any nest elements were either considered inactive (if not moving), or patrolling (if moving). Each of these three within-nest behaviors represents a cluster of previously identified behaviors in bumble bees[61]. "Nursing," for example, incorporates brood thermoregulation, nest construction, larval feeding, and multiple other behaviors associated with brood care; "Patrolling" may include hygienic activity on the nest periphery, as well as patrolling (and potentially transiting between nest structures or out of the nest) and actively buzzing on the nest periphery; "Inactivity" incorporates guarding, resting, perching, and inactivity. We tested repeatability of task allocation proportions within individuals across experimental days via one-way ANOVAs, after accounting for the effects of body mass, colony, and experimental day with a linear mixed effects model[76]. We tested the accuracy of this automated behavioral classification against a human observer using a set of 841 behavioral sequences manually classified by a human observer (Supplementary Table 2).

**Division of labor metrics**. We calculated the degree to which individual workers specialized on a subset of colony tasks ($DOL_{ind}$) using Gorelick's normalized mutual entropy metrics[64] (as updated in refs. [77,78]) using data on the proportion of time each worker spent on different colony tasks (Fig. 1). This metric was first calculated for each colony on separate days, and subsequently calculated for each colony pooling behavioral data across days, either including or excluding data collected overnight (i.e., between 8 p.m. and 6 a.m.).

**Distribution of colony-wide foraging activity**. We quantified the inequality in foraging activity among workers within each colony by calculating the Gini coefficient for foraging transits across all individuals tracked within the nest on days 3–5 after colony deployment. Gini coefficients were calculated in R using the Lorenz curve of foraging activity (i.e., cumulative proportion of foraging activity plotted against an individual bee's foraging activity rank, Fig. 2a). As above in the case of spatial occupancy, the Gini coefficient is calculated by taking the area above the observed Lorenz curve and below the line of perfect equality, relative to the total area under the equality line. Higher Gini coefficients thus reflect more skewed, unequal distributions in the data set. Data from five colonies were removed from this analysis because of insufficient foraging data across these particular days (due to computer or camera failure), but inclusion of partial data for these colonies did not change any qualitative results.

To estimate whether observed values were higher than random expectation, we generated simulated Gini coefficients based on two scenarios for each colony. In the first scenario, total foraging activity was randomly distributed across all individuals equally (1-group sim, Fig. 2c), and in the second, foraging activity was randomly distributed among bees that were actively foraging this time period (2-group sim, Fig. 2c). We generated 100 simulations under each colony for each scenario, then calculated the average Gini coefficient under each simulation condition for each colony.

**Experimental disturbance of foraging**. To examine colony responses to disturbance, we removed foragers from colonies by monitoring outside the nest entrances and collecting up to the first 15 foragers entering or exiting the nest. This technique selects non-randomly for more active foragers and is likely to simulate the effects of heavy natural predation, since foragers would be at risk in proportion to the amount of time they spend foraging[8]. We quantified colony-level foraging metrics (portion of bees foraging, and Gini coefficient) for the three days before and after disturbance (Fig. 2). Tagged bees that were not tracked on any particular day were removed from analyses for those days to avoid bias from tag loss. We also generated expectations for what these colony foraging metrics would be after experimental manipulations based on the loss of the particular foragers that were removed, given the null assumption that the remaining bees in the colony would display no change in their previous foraging behavior.

**High-dimensional nest behavior data**. We used the threshold in movement speed (described above) to calculate the proportion of time each bee was mobile,

separately at night (8 p.m.–6 a.m.; $P_{active,night}$) and during the day (6 a.m.–8 p.m.; $P_{active,day}$). Circadian activity scope ($Circ_{str}$) was calculated as the difference between these two metrics within a given day, yielding a larger score for bees that were more active during the day than at night. Mean moving speed ($Speed_{mov}$) and the standard deviation of moving speed ($Dev_{speed}$) were calculated across all timesteps when the bee was moving (combined across day and night), rather than stationary. We also calculated longer-term movement patterns within the nest by measuring the mean displacement between video recordings ($Disp_{bw,ts}$).

Next, we used the maps of nest components to assess spatial fidelity. For each frame where a bee's spatial position within the nest could be identified, we first calculated the instantaneous distance to each mapped nest element. Bees were considered to be located on the nest element closest to their position (Supplementary Movie 3). If no nest elements were located within 1 cm (or approximately a worker bee's body length), bees were not considered to be physically associated with any nest elements. If these spatial associations changed for an individual bee within a video, the most common spatial association was assigned to that bee for that timestep. We calculated the proportion of time each bee was physically associated with brood, separately at night ($P_{BR,night}$) and during the day ($P_{BR,day}$).

We characterized the spatial distribution of each bee within the nest in a variety of ways, calculating metrics separately for daylight hours (6 a.m.–8 p.m.) and nighttime hours (8 p.m.–6 a.m.). First, we estimated several metrics of spatial distance from the nest center. We defined the "nest center" for each 24-h day, for each colony, as the mean spatial positions of all coordinates from all bees recorded on that day (as in ref. [79]). For each bee, we then measured the "instantaneous" distance as the mean of all instantaneous distances of that bee to the nest center, during the day ($D_{cent,day}$) and at night ($D_{cent,night}$). In addition we separately measured the minimum (nearest) and the median distance to all brood ($D_{BR,nearest}$, $D_{BR,all}$) and food (wax) pots ($D_{WP,nearest}$, $D_{WP,all}$) within the nest.

Next, we calculated a daily spatial probability distribution for each bee by binning all spatial coordinates for a particular bee on a given day (pooling day and night) into a 2-cm grid, normalizing, and smoothing using kernel density estimation (Fig. 3a). In addition, we estimated the similarity between spatial probability distributions both (a) across individuals and (b) within individuals across days, by calculating the correlation of occupancy across all spatial bins (from hereon "spatial correlation").

We used these spatial correlations to calculate pairwise social interaction strengths among all colony members for each day, as this metric incorporates both direct physical interactions, as well as indirect, substrate-based (i.e., stigmergic) interactions[49]. To examine the relationship between spatial correlation and physical interactions between workers, we calculated pairwise distances between all worker pairs for each video frame. Workers were considered "interacting" if their tags were located within 1 cm (approximately a body length) of each other. For each worker, we calculated the number of unique nestmates this focal bee physically interacted with during a single day. We then calculated a mean interaction strength ("Spatial Correlation Strength" in Fig. 3d) for each worker by averaging the social interaction strengths of each worker to all other workers that were present in the colony that day (Fig. 3c) and had sufficient tracking data (>100 individual frames tracked, a condition met by 88% of tracked bee-days). We calculated mean spatial correlation strengths of each bee to all other nestmates ($Cor_{spat,all}$, equivalent to "social interaction strength"), as well as to foragers ($Cor_{spat,frgr}$) and to non-foragers ($Cor_{spat,nfrgr}$) separately, with foragers defined as workers detected at least twice in the foraging tunnel on that particular day.

We then calculated two metrics of the spatial distributions of individual workers for each day: (1) a spatial dispersion index ($Disp_{nest,occ}$; the variance:mean ratio, with higher values associated with a more "clumped" distribution, a commonly used metric in spatial ecology[80]) and (2) a Gini coefficient of spatial occupancy ($Gini_{nest,occ}$). The Gini coefficient scales between 0 and 1, with higher values associated with a more skewed, unequal distribution, and is calculated by taking the relative area above the Lorenz curve (see below for a generic description of the Lorenz curve in the context of foraging activity). We calculated the Gini coefficient for each bee's spatial occupancy separately for each day.

We also calculated a nest area "home range" for each bee. Specifically, we identified the minimum number of grid cells that accounted for 50 and 90% of each bee's occupancy, and then calculated the area of the minimum convex polygon encompassing these grid cells to determine the 50% ($RA_{50}$) and 90% ($RA_{90}$) home range areas, respectively.

For each bee on each day, we also calculated interaction rates with waxpots ($R_{int,WP}$) and brood ($R_{int,BR}$) by multiplying spatial probability distributions by the number of identified nest elements in each spatial bin within the nest. Finally, we measured proportion of time spent on the brood vs. on waxpots ($P_{brood}$) for each worker on each day by calculating the proportion of total time spent on the nest structure that the worker was on the brood (rather than wax or food pots), separately for day and night periods as above.

Nine variables were $\log_{10}$-transformed ($D_{BR,nearest}$, $D_{BR,all}$, $D_{WP,nearest}$, $D_{WP,all}$, $D_{cent,day}$, $D_{cent,night}$, $Speed_{mov}$, $Dev_{speed}$, and $Disp_{bw,ts}$) and four variables were square-root transformed ($R_{int,WP}$, $R_{int,BR}$, $RA_{50}$, and $RA_{90}$).

**Principal components analysis**. To reduce the dimensionality of nest behavior metrics and examine the correlation structure between components of nest behavior, we performed a principal components analysis on daily averages of all estimated

metrics of nest behavior (Fig. 4, Supplementary Table 1). Data were scaled and centered to reduce bias from metrics on different quantitative scales. We tested for significant repeatability of individual principal component scores across days by performing a one-way ANOVA on residual PC1 and PC2 scores (after accounting for effects of mass, colony, and experimental day with a Linear Mixed Effects model). For this and all other analyses of nest behavior, we removed data from bees on days that had less than 40 total timesteps (20 at night and 20 during the day) to assess nest behavior. This quality filter (which removed ~30% of bee-day observations) is unlikely to introduce bias into our analyses, since there was only a weak relationship between number of observations and components of nest behavior (Supplementary Fig. 6), and qualitative results were unchanged if all data were included.

To explicitly examine the relationship between each principal component and relevant information cues within the nest, we separately examined the correlation between principal component scores and three of the above nest behavior metrics known to correlate with information cues within bumble bee colonies; brood interaction rate (a proxy for larval hunger signals[47]), waxpot interaction rate (a proxy for information transferred through food stores[18], and spatial correlation strength (a proxy for both physical[63,81] and substrate-based[49] social interactions). Specifically, we calculated correlation strengths between residual principal component scores and residual scores for each of these information metrics, based on Linear Mixed Effects Models including colony as a random effect and mass as a fixed effect (Supplementary Fig. 4).

We examined the relative importance of several variables (body mass, individual, colony, experimental location, and day from the beginning of the experiment) for explaining variation in the first two principal components scores using hierarchical partitioning[82], as implemented in "hier.part" R package[82,83].

**Forager removal and nest behavior**. To investigate the relationship between nest behavior and foraging activity, we built a series of generalized linear mixed effects models using the "lmer" function in the lme4 package[79] in R[84]. First, we tested the effect of nest behavior on probability of foraging in undisturbed colonies (i.e., all colonies before any experimental manipulations, Fig. 5), with PC1, PC2, and body mass as fixed effects, and colony and individual as random effects. Next, we tested the effect of nest behavior during the three days before simulated foraging on the probability of switching to foraging the day after disturbance, among bees that were previously not foraging (Fig. 6), again with PC1, PC2, and body mass as fixed effects, and colony as a random effect. We built similar models to test the effects of independent metrics of nest behavior on initiating foraging, with $R_{int,WP}$, $R_{int,BR}$, $Cor_{spat,all}$, $P_{act,night}$, $Disp_{bw,ts}$, and mass as fixed effects, and colony as a random effect (Fig. 7). Finally, we tested the effects of switching to foraging after forager removal on changes in nest behavior metrics with separate linear mixed effects models, with foraging activity (binary) as a fixed effect and colony as a random effect (Fig. 8). P values of fixed effects for all models were calculated using the lmerTest[85] package in R.

**Data availability**. Data generated by and presented in this manuscript are available on Zenodo (DOI 10.5281/zenodo.1172834). Custom scripts for behavioral tracking and analysis are available upon request from the corresponding author.

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

## Acknowledgements

This work was support by a NSF GRFP grant to J.D.C., a James S. McDonnell Fellowship to N.G., NSF grant IOS-1253677 to S.A.C., and NSF Grant IOS-1257543 to N.E.P.

## Author contributions

J.D.C., S.D.K., N.E.P. and S.A.C. designed the study. J.D.C., A.M.M. and N.G. designed and built experimental arenas. J.D.C. and R.L.O. performed experiments. J.D.C analyzed and interpreted results. All authors discussed results and contributed to manuscript preparation.

## Additional information

**Competing interests:** The authors declare no competing interests.

