## [Peer Review File(PDF 426 kb) · Nature Communications]

Reviewers' comments:

Reviewer #1 (Remarks to the Author):

This study reports on an analysis of a very rich and interesting set of observations of bumblebee behavior before and after foragers were removed. These data make it possible to address empirically the relation among many different factors that affect foraging behavior which have previously been considered separately. I have some suggestions for revisions that would make the context, data analysis and results more easily understood.

1. On the introduction and literature review:

I applaud the authors' effort to make a conceptual review of the literature, rather than just assuming a starting point. But you might want to consider some changes that would make the straw men a little lighter. For example, I think it is unlikely that anyone really believes that all workers are alike. And when people write papers that discuss factor x rather than factor y, it is most likely because they haven't thought about factor y, not because they believe that factor y does not exist. E.g. it's not that response threshold models require the belief that all individuals are homogenous in space and time; rather, the models tend to ignore this heterogeneity. That's not exactly the same thing.

2. On the methods and interpretation of the image analysis:

a) Line 151-3 "The time each bee spent foraging was estimated by combining information on foraging transits and presence/absence within the colony."

This would be worth explaining more since it is central to the results. If a bee was seen going out and then it's not seen inside the nest, it is recorded as foraging? But sometimes a bee can get in or out without being seen, and sometimes it is in the nest but not seen? How is the measurement error in this estimate combined with the measurement error that leads to 86% agreement between the observers and the image analysis (STable 2c? Then, can you give some relationship between this envelope around the results due to measurement error and the significance levels of the correlations? I realize this is complicated because the significance levels are built around variables that came from a principal components analysis (see below), but some estimate of this would be helpful.

b) Is 'disturbance' on line 266 the same as forager removal which is the same as 'simulated predation'? How closely do forager removals simulate predation; do predators usually show up at the nest entrance and remove bees in the way that the researchers did?

c) The phrase 'portion of time' seems odd - is there a reason not to say 'proportion'?

3. On the data analysis and results :

The analysis tests for relationships among the axes determined by a principal components analysis. It's not clear why the 3 sets of comparisons have the names "correlational", "predictive", and "response threshold". Could any of the 3 be given any of those names?

It would be helpful to give the reader more intuition for how to interpret the results because they involve PC axes rather than variables that were measured directly. For example, if the PC axis that is called 'spatial centrality' really is spatial centrality, then how do you reconcile the result on lines 264-65 that more central bees are more likely to switch to foraging in response to forager removal (is this correct?), with the result on lines 272-273 that foragers are not likely to be centrally located? Or is the latter result (272-273) that bees do not become less centrally located after they start foraging? Does this mean that a bee that is in an interaction hotspot is likely to switch to foraging, but foragers that just switched to foraging tend not to go back to the place they came from? Or perhaps the two results are not related because they are not about the same measure of spatial centrality after all?

More generally, why did you decide not to analyze the data directly by looking at relationships between, e.g. activity level, spatial centrality, and probability of foraging? - rather than the relationships between PC axes that are combinations of those variables? For example, is interaction strength or 'spatial correlation strength' directly associated with the probability that a bee becomes a forager? Is this different before and after the removal of foragers? It would be helpful to give some explanation for why this indirect method was chosen for data analysis, or explain how the results would be different if the variables were compared directly.

It's very interesting - though I don't think surprising- that social interaction is correlated with spatial location and with brood and waxpots (lines 303-305). But it's hard to interpret this result without knowing more about what 'social interaction' and 'spatial location', etc, mean here.

One result seems obvious but perhaps this is again because the variables are not exactly what they sound like they are. For a bee to be measured as foraging it has to move to the entrance and leave the nest, so is there any way that foraging and activity level could not be correlated? If I understand Fig S5 correctly, activity is very bimodal, so bees that are going out to forage are going to be among the ones moving quickly and thus have a higher activity level.

Reviewer #2 (Remarks to the Author):

Overall, interesting dataset. The paper needs improvement in reasoning and references to literature, and I would think it would be more suitable to a specialized journal.

Main suggestions for improvement:

- focus the entire paper more on actual novel questions/hypotheses tested, and delegate the confirmation of known patterns to a subsection with an appropriate heading.
- emphasize what, if anything, is different about your data from previous studies on the same questions. E.g. even on the known pattern of weak task specialization, what information is added by having much more comprehensive data? If nothing, you should point out explicitly that this study now proves that behaviorists need not do continuous monitoring. That is interesting information. If something, point out what that something is. For example, what if anything do we learn from the exact shapes of the task distributions in Fig. 1? Why is nursing linear and others apparently exponential? Do we see more variation in foraging than other tasks?
- to me it seems the most interesting novel aspect is the data on task switching. Who switches when and why? What predicts who switches? With your more comprehensive dataset, you might be able to give better answers to this question than previous studies. Is it spatial position or interaction rate just before the switch, or over the long term for that individual, that predicted the switch event? That distinction would be really interesting. Or is it something about colony dynamics overall?
- make sure you explicitly develop an alternative hypothesis for every hypothesis you test, and point out how you have disproven that alternative. Only by disproving something interesting and plausible does your paper gain interest.
- I make some suggestions below regarding references: you should cite papers containing actual evidence for the statements, not other papers or books who have proposed something without evidence
- it seems you also monitored the locations of brood, honeypots and such? Why not show, analyze, and think more about that data as well? E.g. how evenly is brood distributed, how patchy are their developmental stages, how dynamic over time?
- all results should be compared more explicitly, even in some cases quantitatively, to prior literature on ants and bees. For example, what have other studies shown that removed active workers? Did they or didn't they show replacement, and how was that demonstrated? There are a bunch of such papers.

Major caveats that need to be more explicitly addressed:

- you are NOT monitoring task: you are assuming task based on location. That means that when you refer to 'switching' you are not actually looking at task switching but at movement between locations. Yes there is strong correlation between location and task, but if you substitute one for the other you obviously cannot actually conclude anything about the relationship between task and spatial position, and your measures of spatial fidelity and task fidelity are actually measuring the same thing. What you call spatial fidelity later in the manuscript is not exactly the same, but since both measures are about space, it is a measure of how much spatial patterns at different scales correlate: i.e. is the small-scale pattern of brood distribution across the nest correlated with the larger-scale pattern of bees' spatial fidelity zones?
- saying that the forager distribution is actively regulated implies not only that it is indistinguishable after removals, but that the slow random changes in foraging frequency that happen over time are insufficient to explain the new distribution. Are they?
- it looks like you did not do a random removal control, right? Why not? Generally a 'sham' manipulation of some kind is an important way to check that the effect you see is generated by the crucial part of your manipulation. Here, what if the generally reduced colony size with same brood number required all bees to forage more? Is that really 'replacing' foragers? Or perhaps the disruption just led to more bees leaving the nest (since you didn't measure whether they actually foraged). This needs to be discussed.
- especially given the numbered reference format, you need to point out much more explicitly which references are the ones that actually contain answers to the same questions you are investigating here, and whether this is for the same species or not.

Detailed comments, somewhat in order of reading:

This study certainly contains a fascinating large dataset, somewhat similar to the Mersch et al. paper referenced, but on bumble bees which are harder to monitor. The technique is likely to yield novel insights. This paper is a step towards moving animal behavior into the realm of data-driven science. As with many such papers, however, I think the actual intellectual contribution is not fully realized, and I worry about 'data-driven' coming to mean 'data-rich but hypothesis-poor'. The abstract states for example that this paper shows high behavioral flexibility, individual consistency in task performances, and spatial fidelity - all of which had been shown in the exact same species by references 67 and 37 previously. Foraging activity in particular is skewed, which has been observed pretty much by every paper on bumble bees; it is interesting that this paper shows that the distribution of foragers vs non-foragers is regulated, but not exactly shocking given this has been shown in other social insects and is anecdotally known by everyone who works on bumble bee learning and such (this is why foragers who have undergone the assay are usually removed, to entice new foragers to be designated).

The most interesting and novel aspect referred to in the abstract is the tendency of task switching and its relationship with spatial fidelity. Given that this is essentially the only novel aspect touched upon in the abstract, this should be given more space: i.e. the reader would like to see the details in the abstract on the directionality of this effect, what kind of disturbance is meant, etc.

There is a reason scientists/philosophers of science harp on about 'alternative hypotheses'. The reason is that it is often easy to find tons of data that are consistent with your favorite hypothesis, or even a bunch of apparently contradicting hypotheses. What you need to really increase our knowledge, and to rigorously test a hypothesis, is to point out what could have happened that did NOT happen. I.e. your data are consistent with the idea that spatial location drives task, but what would have happened if that were not the case? What if task is driven purely by genetics, or if spatial fidelity is driven by interactions or by task, or whatever the interesting alternative is? What would these have predicted that did not happen? Without explicitly pointing that out, the information gain from the fact that your data are consistent with some of your hypotheses is minimal, and certainly does not constitute a

'test'.

While I largely agree with the statements made in the general introduction, too many of the references are to reviews, or worse, speculative papers/books. E.g. is colony efficiency increased by morphologically distinct workers who are more efficient and/or by reducing costs associated with task switching? Oster & Wilson do not provide data on either point (yes they show that if these points are true then that would make division of labor a good thing - one might argue we knew this since Adam Smith...). There is newer work on both of those points (ie. work containing empirical evidence). (Yes I agree ref 4 (from 1968) is appropriate here but hardly provides the reader with the current state of the literature)

Or, in line 86 you refer to the claim that individual workers vary in response thresholds and that this determines task performance, but the paper you cite is a conceptual/review paper. The main empirical evidence I am aware of is your reference 34, but that same author has now published work (Westhus et al., Behavioural plasticity in the fanning response of bumblebee workers: impact of experience and rate of temperature change) that seems to suggest that the conclusions from the first paper(s) were misleading, and it is not individually consistent differences, but individually differing recent experience that drives task specialization. Do you disagree? In my view, the task threshold idea, while elegant, is lacking the widespread empirical support one would expect for such a far-reaching phenomenon. I'm not saying you shouldn't mention it, but it is more a hypothesis than an established mechanism at this point.

Or why is Heinrich's book cited for body size variation driving task specialization? I don't have it handy but I'm pretty sure he never measured task specialization inside the nest, so if anything he may have referred to foragers being larger.

Another example: What is it supposed to mean that 'clear polyethism is absent from the vast majority of social insect species'? I'm not aware of a single study that demonstrates that polyethism, defined as differences in behavioral profile across workers, is absent from any species of social insect. In the context as written, it sounds as if you are equating 'polyethism' with 'discrete castes', which is not how it is generally used; moreover, honey bees do not have 'discrete castes' (yes they have age polyethism, but there is considerable overlap between tasks in age distribution, for example - which you yourself point out where ref 10 is cited; and age groups are not 'morphologically specialized', somewhat implied by the wording of the first paragraph).

Minor problem with wording: line 61ff, 'emphases' are 'not exclusive': yes they are, an author can either emphasize one or the other. But you mean that the two actual processes are not mutually exclusive.

Line 72: Individual behavioral variation CAN arise in the absence of genetic variation - almost certainly a lot of the variation we do in fact see in social insects is at least partly genetic.

Line 102: If it's called 'foraging for work model', it assumes that task allocation is a consequence not a cause of spatial worker distribution. You don't give a reference for an example that contradicts this statement. Some of the references for 'foraging for work models' are not models at all (e.g. 42). Other than the work of Sendova-Franks (eg your ref 56, interestingly not cited when talking about task allocation driven by spatial pattern), the main person who has investigated spatial fidelity in social insect workers is Walter Tschinkel, who should be referenced here. The Mersch reference, while certainly interesting, does not give information on the causal direction between task and spatial pattern (and of course neither does yours).

Line 172: consistent with previous findings in bumble bees! Cite those studies, e.g. Cameron or Goulson.

Line 174: To find if this is 'strong skew' one would expect a comparison either to other tasks or to other species. We know it is strong compared to random, that's why it is called task specialization (which had already been demonstrated).

Fig. 2: It's unclear to me what is plotted in A. Are the axes the same? What is rank based on? What does 'portion' mean, should it be 'proportion'?

Fig. 3: For C and D, what do these mean? I.e. what do you now conclude about how bumble bees allocate themselves across space, and how this affects interactions?

Fig. 4: If the 'axes' are 'correlated', what makes them 'axes'? Usually the idea is that we call something an 'axis' when it is orthogonal (thus by definition uncorrelated) to other axes. PC1 and PC2 are orthogonal (thus not correlated). Many of the behavioral metrics correlate with each other. What do we learn from the fact that individual worker behavior differs along arbitrary axes composed of 15 different measured? I'm not saying we learn nothing but I am saying it is your obligation to say what we do learn (and explain it explicitly to the reader).

Also, I wouldn't phrase the 'loading' of a measure on the PC as 'correlation': the PC is calculated using the measures you actually measured - the 'correlation' is thus not the new discovery of a relationship, instead it is a sign that this measure contributes more or less to the differences between individuals. (e.g. line 234)

Line 269: I don't particularly like simply substituting 'activity' for PC2 given that you also directly measured 'inactivity'. Now, in the text, 'activity' is a wholly different measure than 'inactivity', and the two have no relationship. This is confusing for a reader. Why not directly test if 'inactivity' increases for new foragers? And isn't what you are measuring simply the fact that by definition bees that are foraging more have to do less stuff inside the nest? Or is the proportion of time spent on active tasks in the nest (as opposed to be inactive in the nest) higher in new foragers, indicating that when they are inside, they need to rest? If that were the case it would be an interesting result in itself.

Line 339: Please cite which studies show discrete distributions (I assume this implies for task profiles). I know of none.

Line 332ff: ref 42 is not about ants, so probably misnumbered. Either way I don't believe these papers show any 'discrete' associations, if that is supposed to mean discrete, non-overlapping, distinct task groups or spatial groups.

Line 342: This is the first time an actual argument is made that connects concrete results from this study to a previously untested hypothesis. This needs to be given much higher prominence, both in the results themselves and earlier in the discussion. And it has to be argued much more clearly why the fact that task was not actually directly measured should not matter to the strength of this argument.

Line 366: why do the authors believe that alloethism means morphological castes? Alloethism just means 'other behavior' i.e. differences between individuals in behavior.

Given the comments above, I didn't actually read the details of the methods.

Finally, I believe bumblebee should be spelled bumble bee: animals who are actually X are typically spelled with a space before the X, whereas animals who are not actually X don't get a space: cuttlefish not cuttle fish, but honey bee not honeybee.

Some (hopefully) helpful references:

Powell, S., and W. R. Tschinkel. Ritualized conflict in *Odontomachus brunneus* and the generation of interaction-based task allocation: a new organizational mechanism in ants. *Anim. Behav.* 58: 965-972 (1999).

Tschinkel, W. R. Sociometry and sociogenesis of colonies of the harvester ant, *Pogonomyrmex badius*: distribution of workers, brood and seeds within the nest in relation to colony size and season. *Ecol. Entomol.* 24:222-237 (1999).

Leighton G, Charbonneau D, Dornhaus A 2017 'Task-switching is associated with temporal delays in *Temnothorax rugatulus* ants', *Behavioral Ecology* 28: 319-327

Russell et al. 2017 Patterns of pollen and nectar foraging specialization by bumblebees over multiple timescales using RFID, *Sci Rep* 7

Reviewer #1 (Remarks to the Author):

This study reports on an analysis of a very rich and interesting set of observations of bumblebee behavior before and after foragers were removed. These data make it possible to address empirically the relation among many different factors that affect foraging behavior which have previously been considered separately. I have some suggestions for revisions that would make the context, data analysis and results more easily understood.

We thank the reviewer for her/his comments, which have led to a significantly improved manuscript.

1. On the introduction and literature review:

I applaud the authors' effort to make a conceptual review of the literature, rather than just assuming a starting point. But you might want to consider some changes that would make the straw men a little lighter. For example, I think it is unlikely that anyone really believes that all workers are alike. And when people write papers that discuss factor x rather than factor y, it is most likely because they haven't thought about factor y, not because they believe that factor y does not exist. E.g. it's not that response threshold models require the belief that all individuals are homogenous in space and time; rather, the models tend to ignore this heterogeneity. That's not exactly the same thing.

We have substantially restructured and rewritten the Introduction in response to this and Reviewer 2's comments. Specifically, we have continued to cite significant previous work on worker castes (we have removed the term "discrete" throughout), but focused our discussion on identifying gaps in our knowledge about individuals within a colony and their respective behaviors. In terms of our discussion of response threshold models, we agree completely, and have edited this section of the manuscript to convey this point more accurately.

2. On the methods and interpretation of the image analysis:

a) Line 151-3 "The time each bee spent foraging was estimated by combining information on foraging transits and presence/absence within the colony."

This would be worth explaining more since it is central to the results. If a bee was seen going out and then it's not seen inside the nest, it is recorded as foraging? But sometimes a bee can get in or out without being seen, and sometimes it is in the nest but not seen?

We have clarified our description of this method in the Results section. Measurements of foraging activity were based almost entirely on whether a worker exited or entered the nest through the foraging chamber. We used presence within the nest only to correct for instances when, indeed, the bee must have transited through the foraging tunnel without being seen (which we now also address more explicitly in the Methods section). The 86% agreement against a human observer refers specifically to categorizing the three nest behaviors, and we have edited the manuscript to clarify this point.

How is the measurement error in this estimate combined with the measurement error that leads to 86% agreement between the observers and the image analysis (STable 2c? Then, can you give some relationship between this envelope around the results due to measurement error and the significance levels of the correlations? I realize this is complicated because the significance levels are built around variables that came from a principal components analysis (see below), but some estimate of this would be helpful.

The accuracy of task estimation is not incorporated into any subsequent models (i.e. after Figure 1) in the paper, because those subsequent analyses are independent of this task categorization, and instead are performed on underlying metrics of nest behavior (e.g. for principal components analysis). We have restructured the Results section with subheaders that we believe make this point more explicit and easier to follow.

b) Is 'disturbance' on line 266 the same as forager removal which is the same as 'simulated predation'? How closely do forager removals simulate predation; do predators usually show up at the nest entrance and remove bees in the way that the researchers did?

Yes, we meant the same by these two phrases. For clarity, we have removed the term “simulated predation,” from the manuscript, substituting either “forager removal” or “disturbance” for shorthand. We now more explicitly address the relevance of this method for replicating naturalistic predation patterns in the Methods section.

c) The phrase 'portion of time' seems odd - is there a reason not to say 'proportion'?

This has been changed here and throughout the manuscript

3. On the data analysis and results :

The analysis tests for relationships among the axes determined by a principal components analysis.

It's not clear why the 3 sets of comparisons have the names "correlational", "predictive", and "response threshold". Could any of the 3 be given any of those names?

We have separated these sets of results into different figures (now Figures 5,6, and 8) and restructured the text of the Results section to clarify the differences between these separate models. In brief, we specifically mean to separate (1) correlations between worker behavior in the nest and foraging status in colonies before any disruption of the foraging workforce has occurred (“Correlational model”), (2) the role of worker behavior *before* disturbance in predicting which workers initiate foraging in response to disturbance (“Predictive model”), and (3) the effect of initiating foraging on subsequent changes in behavior within the nest (“Response model”). In addition to separating these results into separate figures, we have expanded this section of the Results to clarify these separate analyses and removed the specific references to “Correlational”, “Predictive”, and “Response.”

It would be helpful to give the reader more intuition for how to interpret the results because they involve PC axes rather than variables that were measured directly. For example, if the PC axis that is called 'spatial centrality' really is spatial centrality, then how do you reconcile the result on lines 264-65 that more central bees are more likely to switch to foraging in response to forager removal (is this correct?), with the result on lines 272-273 that foragers are not likely to be centrally located? Or is the latter result (272-273) that bees do not become less centrally located after they start foraging? Does this mean that a bee that is in an interaction hotspot is likely to switch to foraging, but foragers that just switched to foraging tend not to go back to the place they came from? Or perhaps the two results are not related because they are not about the same measure of spatial centrality after all?

The latter description is correct – this result means that more spatially central bees are more likely to become foragers after disturbance, but that those that switch to foraging

do not shift their spatial location to become more central after they begin foraging. We have edited the wording here for clarity, and we believe this is addressed in part by our changes above, which hopefully make the interpretation of these results clearer. In addition, we have expanded the description of these principal component values to aid in interpretation, and added a figure (Fig 7) that separately tests behavioral metrics underlying principal components. We believe these changes will help clarify this section of the results.

More generally, why did you decide not to analyze the data directly by looking at relationships between, e.g. activity level, spatial centrality, and probability of foraging? - rather than the relationships between PC axes that are combinations of those variables? For example, is interaction strength or 'spatial correlation strength' directly associated with the probability that a bee becomes a forager? Is this different before and after the removal of foragers? It would be helpful to give some explanation for why this indirect method was chosen for data analysis, or explain how the results would be different if the variables were compared directly.

It's very interesting - though I don't think surprising- that social interaction is correlated with spatial location and with brood and waxpots (lines 303-305). But it's hard to interpret this result without knowing more about what 'social interaction' and 'spatial location', etc, mean here.

To clarify, foraging activity is not directly included in our principal component analysis – the behavioral metrics included in the principal components analysis focused exclusively on worker behavior within the nest. We have edited the Results section to make this point more explicit. “Spatial centrality” and “activity” are themselves the principal component values, and are now referred to as “Spatial Centrality PC” and “Locomotor Activity PC” for clarity in the revised manuscript.

We chose a principal components approach for two reasons: (1) dimensionality reduction allows for statistically tractable description of a multidimensional dataset (for example, the analyses in Figure 4), and (2) relatedly, many of our analyses are not possible for 23 metrics evaluated simultaneously, especially for analyses performed on smaller sample sizes (e.g. in Figures 6 and 8 in the revised manuscript). To address the reviewer’s concerns, however, we have included a new figure (Fig 7, as described above) that explicitly tests which of the components of “spatial centrality PC”-associated metrics drive individuals to switch tasks. We have also included a supplemental figure (Fig S3) that addresses the same issue for changes in activity (“Locomotor Activity PC” in the revised manuscript) and spatial centrality after initiating foraging and included a verbal description of these results in the Results section.

One result seems obvious but perhaps this is again because the variables are not exactly what they sound like they are. For a bee to be measured as foraging it has to move to the entrance and leave the nest, so is there any way that foraging and activity level could not be correlated? If I understand Fig S5 correctly, activity is very bimodal, so bees that are going out to forage are going to be among the ones moving quickly and thus have a higher activity level.

The time bees take to move to the foraging chamber is generally short, and we believe unlikely to have a strong quantitative effects on our results, but there are several aspects of the data we present here suggesting that the correlation between activity and foraging is not simply driven by this necessary link. First, foraging is negatively correlated with activity level (i.e. foragers tend to be less active when they’re in the nest). Likewise, bees that initiated foraging after disturbance reduced overall activity level, rather than

increasing it (as we would predicted based on this link).

Reviewer #2 (Remarks to the Author):

Overall, interesting dataset. The paper needs improvement in reasoning and references to literature, and I would think it would be more suitable to a specialized journal.

We thank the reviewer for her/his comments, which we believe have led to a significantly improved manuscript. A summary of changes described in more detail below:

- (1) We have substantially rewritten the Introduction and Discussion to focus more on questions of flexible task allocation in social insect colonies and the role of worker spatial fidelity that are directly relevant for the main novel results we present here. Our Introduction now more explicitly explains the motivation for gathering high-throughput, multi-dimensional datasets to achieve a more holistic understanding of behavioral variation among workers (e.g. L102-111, among other places).**
- (2) We have restructured the Results section, including adding subsections for clarity, and new results in Figures 1 and 4, new figures (Figures 7 and S3). We restructured the results previously presented in Figure 5 into three Figures (5,6, and 8) to aid in clarity and emphasis.**
- (3) We have removed more speculative citations (especially in the early Introduction) and added several new citations and discussion of papers directly relevant to our questions surrounding forager replacement and nest behavior throughout.**
- (4) We clearly outline hypotheses and discuss alternatives in both the Introduction and Discussion, specifically focusing on our results surrounding forager removal and replacement, and the role of spatial fidelity in driving particular workers to initiate foraging after disturbance.**

Main suggestions for improvement:

- focus the entire paper more on actual novel questions/hypotheses tested, and delegate the confirmation of known patterns to a subsection with an appropriate heading.

We have substantially edited the Introduction and Discussion to focus more specifically on the novel questions we address in the study, and for each component, clarify which aspects recapitulate known patterns, and which provide novel insights compared to previous studies. This includes explicitly enumerating hypotheses at the end of the Introduction, and adding discussion of alternative hypotheses in the Discussion section. While we have not created a dedicated subsection specifically for novel results, we have introduced subsections for different aspects of the Results, and made edits to text throughout to clarify which components of our results confirm known patterns from previous work, and which extend known patterns with novel insights.

- emphasize what, if anything, is different about your data from previous studies on the same questions. E.g. even on the known pattern of weak task specialization, what information is added by having much more comprehensive data? If nothing, you should point out explicitly that this study now proves that behaviorists need not do continuous monitoring. That is interesting

information. If something, point out what that something is. For example, what if anything do we learn from the exact shapes of the task distributions in Fig. 1? Why is nursing linear and others apparently exponential? Do we see more variation in foraging than other tasks?

Throughout the manuscript, we have incorporated edits that explicitly highlight the novelty of our work relative to the existing literature. In the context of task specialization, we have added figure panels that highlight (1) that our automated task classification system is generally consistent with patterns of worker specialization generated by manual behavioral monitoring (Figure 1G, and discussed explicitly as a validation of known patterns in the revised manuscript), and (2) that continuous tracking adds to our understanding of these patterns, since incorporating overnight data on overnight individual task performance shows that day-only behavioral monitoring systematically overestimates the degree of worker specialization.

- to me it seems the most interesting novel aspect is the data on task switching. Who switches when and why? What predicts who switches? With your more comprehensive dataset, you might be able to give better answers to this question than previous studies. Is it spatial position or interaction rate just before the switch, or over the long term for that individual, that predicted the switch event? That distinction would be really interesting. Or is it something about colony dynamics overall?

We agree that this is the most novel component of our study, and have substantially edited the Introduction, Results, and Discussion to emphasize and expand on these results. In addition to including a new figure (Fig 7) that explicitly addresses which specific components of spatial fidelity predict the workers that switch task after forager removal, we have replaced Figure 5 in the previous manuscript with 3 figures (Figs 5, 6, and 8) and expanded the description of these in the Results section to further highlight these data, and expanded their interpretation significantly in the Discussion section.

- make sure you explicitly develop an alternative hypothesis for every hypothesis you test, and point out how you have disproven that alternative. Only by disproving something interesting and plausible does your paper gain interest.

We have incorporated specific discussion of alternative hypotheses throughout. Two examples include that (1) we now explicitly address alternative hypotheses about which components of spatially-encoded information within bumblebee nests drive patterns of task switching (with new text in the Introduction, a new Figure in the Results, and new text in the Discussion) and (2) we now specifically address alternative hypotheses from the literature on the relationship between mobility and space use in social insect colonies (addressed in Figures 5-8, and in the edited Discussion).

- I make some suggestions below regarding references: you should cite papers containing actual evidence for the statements, not other papers or books who have proposed something without evidence

We have incorporated these specific references (which we thank the reviewer for suggesting) and others, and have removed references to more speculative papers and books (described in more detail below).

- it seems you also monitored the locations of brood, honeypots and such? Why not show,

analyze, and think more about that data as well? E.g. how evenly is brood distributed, how patchy are their developmental stages, how dynamic over time?

We agree that these data are quite interesting. We have made edits to the manuscript that clarify the ways in which these data are incorporated into our analysis (of which there are many, including specifically in the new Figure 7). While we believe a full analysis of all these data is beyond the scope of this particular manuscript given space constraints and the other changes to the revised manuscript, these data will be made fully available for future inquiries on these very interesting questions.

- all results should be compared more explicitly, even in some cases quantitatively, to prior literature on ants and bees. For example, what have other studies shown that removed active workers? Did they or didn't they show replacement, and how was that demonstrated? There are a bunch of such papers.

We have substantially rewritten the Introduction and Discussion to focus more explicitly on task switching and the important questions raised by the reviewer, including several new references to this point. In addition, we have included quantitative data from other studies when appropriate (e.g. in Figure 1G).

A partial list of new references in the revised manuscript relevant to flexible regulation of foraging:

Camazine, S. The regulation of pollen foraging by honey bees: how foragers assess the colony's need for pollen. *Behav Ecol Sociobiol* 32, 265–272 (1993).

Huang, Z.-Y. & Robinson, G. E. Regulation of honey bee division of labor by colony age demography. *Behav Ecol Sociobiol* 39, 147–158 (1996).

Gordon, D. M., Holmes, S. & Nacu, S. The short-term regulation of foraging in harvester ants. *Behavioral Ecology* 19, 217–222 (2007).

Kwapich, C. L. & Tschinkel, W. R. Demography, demand, death, and the seasonal allocation of labor in the Florida harvester ant (*Pogonomyrmex badius*). *Behav Ecol Sociobiol* 67, 2011–2027 (2013).

Major caveats that need to be more explicitly addressed:

- you are NOT monitoring task: you are assuming task based on location. That means that when you refer to 'switching' you are not actually looking at task switching but at movement between locations. Yes there is strong correlation between location and task, but if you substitute one for the other you obviously cannot actually conclude anything about the relationship between task and spatial position, and your measures of spatial fidelity and task fidelity are actually measuring the same thing. What you call spatial fidelity later in the manuscript is not exactly the same, but since both measures are about space, it is a measure of how much spatial patterns at different scales correlate: i.e. is the small-scale pattern of brood distribution across the nest correlated with the larger-scale pattern of bees' spatial fidelity zones?

This is an important point, and we thank the reviewer for this comment. To clarify, we use spatial data on the distribution of workers within the nest in two distinct ways in the manuscript: (1) we use a combination of space-use and activity to provide a simple, automated categorization of task performance (with task broadly defined, as discussed

in the manuscript), and (2) we incorporate a more detailed, multi-dimensional set of metrics related to nest space-use into a principal components analysis to reduce the dimensionality of many aspects of worker behavior within the nest. These two analyses are functionally separate (the task categorization from Figure 1 is not directly incorporated into the principal components analysis, even though the same raw data is used for both analyses).

We have restructured the Results section with subsections and text edits that we believe help to clarify this. The reliability of these spatial data in classifying task category (as in Fig 1) is very important and is directly addressed in the manuscript and Table S1.

For the latter section of the manuscript (E.g. Fig 5 in original manuscript, Figs 5-8 in revised manuscript), our use of “task switching” was limited to whether or not bees were foraging, the measurement of which is entirely independent of spatial distributions of workers within the nest (a point which we have clarified in the Results section of our revised manuscript). Thus, while spatial data within the nest are relevant for these analyses in their contribution to Principal Component scores (and now are independently analyzed in Figure 7 and Figure S3), they do not contribute to the measurement of the relevant task (foraging). We agree that this terminology was confusing, and have removed the phrase “task switching” from this section of the manuscript and replaced it with more explicitly descriptive phrases, such as “initiate foraging”.

- saying that the forager distribution is actively regulated implies not only that it is indistinguishable after removals, but that the slow random changes in foraging frequency that happen over time are insufficient to explain the new distribution. Are they?

Two lines of evidence suggest that the patterns of foraging activity we see after forager removal are not random. First, we show that both the Gini coefficient and the proportion of bees foraging are significantly higher after disturbance than would be expected if the remaining workers maintained their pre-disturbance levels of foraging activity. We would not see a significant difference between observed and expected outcomes in this case if there the changes were slow and random, but only if the shifts in foraging behavior of the workers across colonies were consistently biased (i.e. not random). Second, we did not see similar changes on the same time scale in either of these metrics in colonies where foragers were not removed (Fig S1).

- it looks like you did not do a random removal control, right? Why not? Generally a 'sham' manipulation of some kind is an important way to check that the effect you see is generated by the crucial part of your manipulation. Here, what if the generally reduced colony size with same brood number required all bees to forage more? Is that really 'replacing' foragers? Or perhaps the disruption just led to more bees leaving the nest (since you didn't measure whether they actually foraged). This needs to be discussed.

That is correct, we did not perform a random removal experiment. While we believe the controls and analyses we include in the manuscript are sufficient to support the conclusion that colonies respond actively to the removal of foragers (see comment above), this does not exclude the possibility that there are also responses to other disturbances such as removal of random workers, and we now mention these points in the Discussion. The increase in foraging activity among remaining workers after removal of foragers is unlikely to result from reduced colony size, since in undisturbed colonies,

colony size was not correlated with the proportion of bees within the colony foraging, a result we now include in the paper (L262-264).

- especially given the numbered reference format, you need to point out much more explicitly which references are the ones that actually contain answers to the same questions you are investigating here, and whether this is for the same species or not.

As noted above, we have substantially rewritten the Introduction to more directly highlight literature relevant for the central questions that our manuscript addresses, and where appropriate have noted if these studies were performed in *B. impatiens*. Throughout these sections, we have included more specific discussion of the content of relevant papers to address this issue.

Detailed comments, somewhat in order of reading:

This study certainly contains a fascinating large dataset, somewhat similar to the Mersch et al. paper referenced, but on bumble bees which are harder to monitor. The technique is likely to yield novel insights. This paper is a step towards moving animal behavior into the realm of data-driven science. As with many such papers, however, I think the actual intellectual contribution is not fully realized, and I worry about 'data-driven' coming to mean 'data-rich but hypothesis-poor'. The abstract states for example that this paper shows high behavioral flexibility, individual consistency in task performances, and spatial fidelity - all of which had been shown in the exact same species by references 67 and 37 previously. Foraging activity in particular is skewed, which has been observed pretty much by every paper on bumble bees; it is interesting that this paper shows that the distribution of foragers vs non-foragers is regulated, but not exactly shocking given this has been shown in other social insects and is anecdotally known by everyone who works on bumble bee learning and such (this is why foragers who have undergone the assay are usually removed, to entice new foragers to be designated).

The most interesting and novel aspect referred to in the abstract is the tendency of task switching and its relationship with spatial fidelity. Given that this is essentially the only novel aspect touched upon in the abstract, this should be given more space: i.e. the reader would like to see the details in the abstract on the directionality of this effect, what kind of disturbance is meant, etc.

We thank the reviewer for her/his perspective on our manuscript. We agree that the task switching experiments in our study have generated the most novel results. As described above, we have significantly restructured the Introduction to focus on questions surrounding task switching and components of worker behavioral variation that might play a role in these dynamics, with a particular emphasis on spatial fidelity. We have introduced a new analysis on the specific components of worker nest spatial fidelity that predict which non-foraging workers initiate foraging following disturbance (Fig 7), and we have taken the analyses previously described in Figure 5 and rearranged them into three figures to make the analyses more clear. Thus the the task switching analyses now comprise roughly half of the figures and results in the manuscript, in addition to editing the Results section to highlight and clarify these results. We more specifically enumerate and address hypotheses (and alternatives) in the Introduction and Discussion sections for our key findings. We have added analyses and edited language that specifically highlight how our results expand on previously known patterns from *B. impatiens*, including in the specific cases of behavioral flexibility and spatial fidelity.

There is a reason scientists/philosophers of science harp on about 'alternative hypotheses'. The reason is that it is often easy to find tons of data that are consistent with your favorite hypothesis, or even a bunch of apparently contradicting hypotheses. What you need to really increase our knowledge, and to rigorously test a hypothesis, is to point out what could have happened that did NOT happen. I.e. your data are consistent with the idea that spatial location drives task, but what would have happened if that were not the case? What if task is driven purely by genetics, or if spatial fidelity is driven by interactions or by task, or whatever the interesting alternative is? What would these have predicted that did not happen? Without explicitly pointing that out, the information gain from the fact that your data are consistent with some of your hypotheses is minimal, and certainly does not constitute a 'test'.

We have edited the manuscript to more explicitly introduce the hypotheses tested here, both in the Introduction and the Discussion sections. Two central hypotheses tested in our manuscript are that (a) variation in space-use within nests before disturbance predicts which workers initiate foraging after disturbance, and (b) switching to foraging results in changes in locomotor activity patterns. We now explicitly address alternative hypotheses for the relationship between mobility, space use, and foraging among bumblebee workers. Our experimental design allows us to test these alternative hypotheses (for example, that individual variation in mobility drives worker spatial fidelity and thus task performance), and our revised manuscript now more specifically considers (and rejects) these possibilities.

While I largely agree with the statements made in the general introduction, too many of the references are to reviews, or worse, speculative papers/books. E.g. is colony efficiency increased by morphologically distinct workers who are more efficient and/or by reducing costs associated with task switching? Oster & Wilson do not provide data on either point (yes they show that if these points are true then that would make division of labor a good thing - one might argue we knew this since Adam Smith...). There is newer work on both of those points (ie. work containing empirical evidence).

(Yes I agree ref 4 (from 1968) is appropriate here but hardly provides the reader with the current state of the literature)

We have removed many of these references in our restructured introduction. We retain the reference to Oster and Wilson, but clarify that this work is theoretical. We also add the recent reference for task switching costs as suggested by the reviewer.

Or, in line 86 you refer to the claim that individual workers vary in response thresholds and that this determines task performance, but the paper you cite is a conceptual/review paper. The main empirical evidence I am aware of is your reference 34, but that same author has now published work (Westhus et al., Behavioural plasticity in the fanning response of bumblebee workers: impact of experience and rate of temperature change) that seems to suggest that the conclusions from the first paper(s) were misleading, and it is not individually consistent differences, but individually differing recent experience that drives task specialization. Do you disagree? In my view, the task threshold idea, while elegant, is lacking the widespread empirical support one would expect for such a far-reaching phenomenon. I'm not saying you shouldn't mention it, but it is more a hypothesis than an established mechanism at this point.

We agree with this description, and have edited this section accordingly, as well as introducing this additional reference.

Or why is Heinrich's book cited for body size variation driving task specialization? I don't have it handy but I'm pretty sure he never measured task specialization inside the nest, so if anything he may have referred to foragers being larger.

We have removed this reference

Another example: What is it supposed to mean that 'clear polyethism is absent from the vast majority of social insect species'? I'm not aware of a single study that demonstrates that polyethism, defined as differences in behavioral profile across workers, is absent from any species of social insect. In the context as written, it sounds as if you are equating 'polyethism' with 'discrete castes', which is not how it is generally used; moreover, honey bees do not have 'discrete castes' (yes they have age polyethism, but there is considerable overlap between tasks in age distribution, for example - which you yourself point out where ref 10 is cited; and age groups are not 'morphologically specialized', somewhat implied by the wording of the first paragraph).

This sentence (and the paragraph containing) have been removed from the manuscript for space constraints.

Minor problem with wording: line 61ff, 'emphases' are 'not exclusive': yes they are, an author can either emphasize one or the other. But you mean that the two actual processes are not mutually exclusive.

This sentence has been removed

Line 72: Individual behavioral variation CAN arise in the absence of genetic variation - almost certainly a lot of the variation we do in fact see in social insects is at least partly genetic.

For brevity on this complex issue, we have simply noted here that behavioral variation arises from many sources in social insects, and now refer readers to a thorough recent review

Line 102: If it's called 'foraging for work model', it assumes that task allocation is a consequence not a cause of spatial worker distribution. You don't give a reference for an example that contradicts this statement. Some of the references for 'foraging for work models' are not models at all (e.g. 42).

Other than the work of Sendova-Franks (eg your ref 56, interestingly not cited when talking about task allocation driven by spatial pattern), the main person who has investigated spatial fidelity in social insect workers is Walter Tschinkel, who should be referenced here.

The Mersch reference, while certainly interesting, does not give information on the causal direction between task and spatial pattern (and of course neither does yours).

We agree with this point – in “foraging-for-work” models, space-use is causal in the sense that when workers occupy a particular space, they perform a particular task, but differences in space-use of workers (and thus task) are driven by external factors, for example age (as in ref 56, Tschinkel 1999), aggression (as in ref 60, Powell and Tschinkel 1999), or mobility (as in ref 59, Backen, Sendova-Franks, and Franks 2000). We have edited this paragraph to clarify this point, and included the references that the reviewer suggests.

Line 172: consistent with previous findings in bumble bees! Cite those studies, e.g. Cameron or Goulson.

We have included new citations specifically addressing the individual repeatability of behavior of bumble bees into the Discussion, where this citation and discussion of this point has been moved.

Line 174: To find if this is 'strong skew' one would expect a comparison either to other tasks or to other species. We know it is strong compared to random, that's why it is called task specialization (which had already been demonstrated).

We have removed the word “strong”

Fig. 2: It's unclear to me what is plotted in A. Are the axes the same? What is rank based on? What does 'portion' mean, should it be 'proportion'?

We have changed the axis labels of this panel and the description of this panel in the figure caption to clarify what is being plotted here. We have changed “portion” to “proportion” here and throughout.

Fig. 3: For C and D, what do these mean? I.e. what do you now conclude about how bumble bees allocate themselves across space, and how this affects interactions?

We have added descriptive phrasing in the Results section clarifying that the purpose of these figures is to show (a) that individuals vary in their mean spatial overlap (i.e. spatial correlation strength) to nestmates, and that (b) workers with higher spatial correlation strength scores have more unique physical interactions with nestmates.

Fig. 4: If the 'axes' are 'correlated', what makes them 'axes'? Usually the idea is that we call something an 'axis' when it is orthogonal (thus by definition uncorrelated) to other axes. PC1 and PC2 are orthogonal (thus not correlated). Many of the behavioral metrics correlate with each other. What do we learn from the fact that individual worker behavior differs along arbitrary axes composed of 15 different measured? I'm not saying we learn nothing but I am saying it is your obligation to say what we do learn (and explain it explicitly to the reader).

We have changed the wording in the figure legend to remove the term “axis.” In the Results section, we have more explicitly described the dimensionality reduction techniques used in Figure 4. In addition, we have changed the labels in Figure 4A, and added four panels to this Figure, with an accompanying section in the Discussion section.

Also, I wouldn't phrase the 'loading' of a measure on the PC as 'correlation': the PC is calculated using the measures you actually measured - the 'correlation' is thus not the new discovery of a relationship, instead it is a sign that this measure contributes more or less to the differences between individuals. (e.g. line 234)

We have replaced “correlation” in this context with “loading” to clarify.

Line 269: I don't particularly like simply substituting 'activity' for PC2 given that you also directly measured 'inactivity'. Now, in the text, 'activity' is a wholly different measure than 'inactivity', and

the two have no relationship. This is confusing for a reader. Why not directly test if 'inactivity' increases for new foragers? And isn't what you are measuring simply the fact that by definition bees that are foraging more have to do less stuff inside the nest? Or is the proportion of time spent on active tasks in the nest (as opposed to be inactive in the nest) higher in new foragers, indicating that when they are inside, they need to rest? If that were the case it would be an interesting result in itself.

We agree that this was confusing. We have changed “activity” here and throughout the manuscript for “Locomotor Activity PC” to clarify interpretation. As discussed above, we have also restructured the Results to emphasize the fact that the task performance categories earlier in the manuscript (including “Inactivity”) are not used in the Principal Component analysis. The reason to use Principal Component scores rather than individual variables is that allows investigation of suites of correlated behaviors (e.g. all of the behavioral metrics associated with PC2). We agree, however, that this approach can make interpretation more difficult. To address this, we have included a new supplementary figure (Figure S3) that includes statistical tests of isolated variables (based on hierarchical clustering of original nest behavior variables shown in Figure S2). This figure shows that we see results that are consistent with a general change in PC2-associated metrics when those variables are analyzed separately, including specifically showing that indeed new foragers reduce locomotor activity (i.e. spend a lower proportion of their time moving) overnight.

Line 339: Please cite which studies show discrete distributions (I assume this implies for task profiles). I know of none.

For space constraints given other changes in the manuscript, this paragraph has been removed, and the term “discrete” has been removed in reference to behavioral castes throughout.

Line 332ff: ref 42 is not about ants, so probably misnumbered. Either way I don't believe these papers show any 'discrete' associations, if that is supposed to mean discrete, non-overlapping, distinct task groups or spatial groups.

We thank the reviewer for noticing this; this reference was indeed misnumbered, and we have removed the term “discrete” here and throughout the manuscript.

Line 342: This is the first time an actual argument is made that connects concrete results from this study to a previously untested hypothesis. This needs to be given much higher prominence, both in the results themselves and earlier in the discussion. And it has to be argued much more clearly why the fact that task was not actually directly measured should not matter to the strength of this argument.

As described above, we have greatly expanded our discussion of the relationship between spatial fidelity, activity, and foraging and the novel insights provided by following nest behavior before and after the removal of foragers. In addition, we have added analyses (Fig 7 and Fig S3) showing that analysis of original metrics yields results consistent with analysis of PC scores.

Line 366: why do the authors believe that alloethism means morphological castes? Alloethism just means 'other behavior' i.e. differences between individuals in behavior.

While we agree with the reviewer about the etymological origins of this word, “alloethism” is widely used to refer to differences in behavior associated variation in body size (see a partial list of references using this term below), and thus is a more general term than “morphological castes”, consistent with how we use it in this context. Because of its relevance to discussions of body size in the social insect literature, and in *Bombus* specifically, we have left this term in the paper, but have moved it inside of the parentheses, placing emphasis on the phenomenon rather than the term.

A partial list of references:

Goulson, D. *et al.* Can alloethism in workers of the bumblebee, *Bombus terrestris*, be explained in terms of foraging efficiency? *Animal Behaviour* 64, 123–130 (2002).

Kapustjanskij, A., Streinzer, M., Paulus, H. F. & Spaethe, J. Bigger is better: implications of body size for flight ability under different light conditions and the evolution of alloethism in bumblebees. 21, 1130–1136 (2007).

Wilson, E. O. Caste and division of labor in leaf-cutter ants (Hymenoptera: Formicidae: *Atta*). *Behav Ecol Sociobiol* 7, 157–165 (1980).

Fowler, H. G. Alloethism in the Carpenter Ant, *Camponotus pennsylvanicus* (Hymenoptera: Formicidae). *Entomologia Generalis* 11, 69–76 (1985).

Given the comments above, I didn't actually read the details of the methods.

We appreciate the attention the reviewer has given to the main body of our manuscript, which we believe has led to significant improvements in our revised version. While we believe information contained in the results is important for several of the concerns the reviewer raises above, we recognize the challenge of having important details moved to the end of the manuscript in the Nature Communications format, and have incorporated more of these key details from the methods into the main body of the revised manuscript.

Finally, I believe bumblebee should be spelled bumble bee: animals who are actually X are typically spelled with a space before the X, whereas animals who are not actually X don't get a space: cuttlefish not cuttle fish, but honey bee not honeybee.

We have changed this throughout the manuscript

Some (hopefully) helpful references:

Powell, S., and W. R. Tschinkel. Ritualized conflict in *Odontomachus brunneus* and the generation of interaction-based task allocation: a new organizational mechanism in ants. *Anim. Behav.* 58: 965-972 (1999).

Tschinkel, W. R. Sociometry and sociogenesis of colonies of the harvester ant, *Pogonomyrmex badius*: distribution of workers, brood and seeds within the nest in relation to colony size and season. *Ecol. Entomol.* 24:222-237 (1999).

Leighton G, Charbonneau D, Dornhaus A 2017 'Task-switching is associated with temporal delays in *Temnothorax rugatulus* ants', *Behavioral Ecology* 28: 319-327

Russell et al. 2017 Patterns of pollen and nectar foraging specialization by bumblebees over multiple timescales using RFID, Sci Rep 7

These (and other) new citations have been incorporated into the revised manuscript, and we thank the reviewer for these very helpful suggestions.

REVIEWERS' COMMENTS:

Reviewer #1 (Remarks to the Author):

This is novel and interesting work. The manuscript is much improved and does a much better job of explaining the results and why they are interesting.

I would suggest one more revision to improve the writing. Often the writing is difficult to follow and some editing would help. This is a matter of style not content; not changing what is being said but instead how it is said.

I will give some examples below of ways that the writing could be confusing to a reader. I'd suggest this kind of editing throughout.

E.g. lines 270-73

270-273 While our automated task classification system incorporates data on movement and 271 space-use (including data from both day and night periods, see Methods for details), 272 categorization into such discrete behaviors is unable to capture many components of 273 complex variation between individuals in nest behavior

What does 'such discrete behaviors' refer to?

Why the part in parentheses - could the reader understand this sentence without knowing that data are from both day and night, or could the reader assume that they are? The instruction to see methods for details is distracting because the reader doesn't know why these details are needed here. Could just say 'components of variation' - don't need the 'complex'.

Could you say here more directly what kinds of variation you are interested in here? e.g. the relation of spatial distributions and interaction patterns.

How about just beginning this paragraph with the next sentence, explaining what the data were: 274-75 We thus generated a high- dimensional dataset on behavior within the nest that included metrics of activity level, movement patterns, and spatial distribution measured for each worker on each day.

What is the spatial distribution for each worker?

Could this be stated as:

For each worker on each day, we found metrics of activity level, movement patterns, and spatial distribution (?) inside the nest.

If this is the same dataset introduced on line 299? consider describing it only once.

I'm puzzled by '(excluding foraging activity)', line 301, because there is no foraging activity inside the nest, right? so how was it excluded?

Again on lines 277, the measure is introduced in terms of what some other measure does not provide.

The sentence beginning on 281 'We measured daily individual patterns of space-use...' states an interesting result and should not be buried in the middle of the paragraph.

(287 correlated with, not to)

In the measure of spatial correlation strength described on 289-294, what is a 'unique interaction'

with a nestmate, line 292? - is that an interaction between particular individuals? If a bee has more of them does that mean it interacts with more different individuals, or it interacts with the same ones more often? If the former, I'm puzzled about why that is related to spatial correlation. Overall does this sentence say that what is called 'spatial correlation strength' in Fig 3D is later referred to as 'social interaction strength'? How about giving this just one name and sticking to it?

Again, these are matters of style and with some more attention to the writing, it will be easier for readers to understand why the work is important and interesting.

Reviewer #2 (Remarks to the Author):

I appreciate the effort the authors invested in improving the manuscript, particularly regarding clarifying hypotheses and alternatives and specifying which novel insights have been gained here. I am happy with the revisions and would now recommend the manuscript for publication.

Minor remaining comment:

line 43 is missing 'in': interest in individual...

REVIEWERS' COMMENTS:

Reviewer #1 (Remarks to the Author):

This is novel and interesting work. The manuscript is much improved and does a much better job of explaining the results and why they are interesting.

We thank the Reviewer for her/his comments and this and the previous version of this manuscript, which we believe have led to significant improvement. We have detailed our responses and changes to the manuscript below.

I would suggest one more revision to improve the writing. Often the writing is difficult to follow and some editing would help. This is a matter of style not content; not changing what is being said but instead how it is said.

I will give some examples below of ways that the writing could be confusing to a reader. I'd suggest this kind of editing throughout.

We appreciate the reviewer's comments. In addition to addressing the specific concerns outlined below, we have edited the entire manuscript for clarity and brevity.

E.g. lines 270-73

270-273 While our automated task classification system incorporates data on movement and 271 space-use (including data from both day and night periods, see Methods for details), 272 categorization into such discrete behaviors is unable to capture many components of 273 complex variation between individuals in nest behavior

What does 'such discrete behaviors' refer to?

Why the part in parentheses - could the reader understand this sentence without knowing that data are from both day and night, or could the reader assume that they are? The instruction to see methods for details is distracting because the reader doesn't know why these details are needed here.

Could just say 'components of variation' - don't need the 'complex'.

Could you say here more directly what kinds of variation you are interested in here? e.g. the relation of spatial distributions and interaction patterns.

How about just beginning this paragraph with the next sentence, explaining what the data were:

274-75 We thus generated a high- dimensional dataset on behavior within the nest that included metrics of activity level, movement patterns, and spatial distribution measured for each worker on each day.

We have incorporated this suggestion into the manuscript, removing the earlier sentence, and edited this section significantly.

What is the spatial distribution for each worker?

Could this be stated as:

For each worker on each day, we found metrics of activity level, movement patterns, and spatial distribution (?) inside the nest.

We have significantly edited this section for clarity

If this is the same dataset introduced on line 299? consider describing it only once.

Dataset described in Line 299 is distinct, so we have left this description intact.

I'm puzzled by '(excluding foraging activity)', line 301, because there is no foraging activity inside the nest, right? so how was it excluded?

We have removed the phrase "excluding foraging activity"

Again on lines 277, the measure is introduced in terms of what some other measure does not provide.

We have removed this sentence

The sentence beginning on 281 'We measured daily individual patterns of space-use...' states an interesting result and should not be buried in the middle of the paragraph.

This is now the second sentence of this section

(287 correlated with, not to)

c

changed

In the measure of spatial correlation strength described on 289-294, what is a 'unique interaction' with a nestmate, line 292? - is that an interaction between particular individuals? If a bee has more of them does that mean it interacts with more different individuals, or it interacts with the same ones more often? If the former, I'm puzzled about why that is related to spatial correlation. Overall does this sentence say that what is called 'spatial correlation strength' in Fig 3D is later referred to as 'social interaction strength'? How about giving this just one name and sticking to it?

We have edited this section for clarity. We have retained the initial description of this correlation as “spatial correlation strength,” (since we think it’s important to first establish the validity of using spatial correlations as a proxy for social interactions), but have edited the sentence for clarity.

Again, these are matters of style and with some more attention to the writing, it will be easier for readers to understand why the work is important and interesting.

Reviewer #2 (Remarks to the Author):

I appreciate the effort the authors invested in improving the manuscript, particularly regarding clarifying hypotheses and alternatives and specifying which novel insights have been gained here. I am happy with the revisions and would now recommend the manuscript for publication.

We again thank the reviewer for her/his help input on the manuscript.

Minor remaining comment:

line 43 is missing 'in': interest in individual...

This sentence has been removed from the manuscript